# Stratospheric influence on the winter North Atlantic storm track in subseasonal reforecasts

Hilla Afargan-Gerstman[1], Dominik Büeler[1], C. Ole Wulff[2], Michael Sprenger[1], and Daniela I.V. Domeisen[3,1]

[1]Institute for Atmospheric and Climate Science, ETH Zurich, Zurich, Switzerland
[2]NORCE Norwegian Research Centre, Bjerknes Centre, Bergen, Norway
[3]University of Lausanne, Lausanne, Switzerland

**Correspondence:** Hilla Afargan-Gerstman (hilla.gerstman@env.ethz.ch)

**Abstract.**

Extreme stratospheric polar vortex events, such as sudden stratospheric warmings (SSW) or extremely strong polar vortex events, can have a significant impact on surface weather in winter. SSWs are most often associated with negative North Atlantic Oscillation (NAO) conditions, cold air outbreaks in the Arctic and a southward-shifted extratropical storm track, while strong polar vortex events tend to be followed by a positive phase of the NAO, relatively warm conditions in the extratropics and a poleward-shifted storm track. Such changes in the storm track position and associated extratropical cyclone frequency over the North Atlantic and Europe can increase the risk of extreme windstorm, flooding or heavy snowfall over populated regions. Skillful predictions of the downward impact of stratospheric polar vortex extremes can therefore improve the predictability of extratropical winter storms on subseasonal timescales. However, there exists a strong inter-event variability in these downward impacts on the tropospheric storm track. Using ECMWF reanalysis data and ECMWF reforecasts from the Subseasonal to Seasonal (S2S) Prediction Project database, we investigate the stratospheric influence on extratropical cyclones, identified with a cyclone detection algorithm. Following SSWs, there is an equatorward shift in cyclone frequency over the North Atlantic and Europe in reforecasts, and the opposite response is observed after strong polar vortex events, consistent with the response in reanalysis. However,although 80% of the reforecasts forecast the sign of the cyclone frequency response over the North Atlantic during weeks 1-4 of SSWs with a canonical surface impact (i.e., an equatorward shift of the storm track), less than 25% of the reforecasts capture the response during SSWs with a 'non-canonical' impact, suggesting a possible overconfidence of the reforecasts with respect to reanalysis in predicting the canonical response after SSWs, which occurs in only two thirds of the events. The cyclone forecasts following strong polar vortex events are generally more successful. Understanding the role of the stratosphere in subseasonal variability and predictability of storm tracks during winter can provide a key for reliable forecasts of midlatitude storms and their surface impacts.

## 1 Introduction

Extratropical cyclones along the North Atlantic storm track have a strong impact on regional weather and climate in Europe, giving rise to extreme weather hazards such as heavy precipitation and strong surface winds. These storms typically develop and

intensify over the baroclinic regions in the western part of the North Atlantic, where strong meridional temperature gradients
are found. In midlatitudes, the position of the storm track, i.e., the aggregated paths of extratropical cyclones, is closely related
to the jet stream, and is often found on the poleward flank of the jet (e.g., Blackmon et al., 1977; Chang et al., 2002; Shaw et al.,
2016). In the North Atlantic, the occurrence of intense extratropical cyclones can produce extreme surface winds, leading in
some cases to severe damage over Europe, huge economic losses and even casualties (e.g., Befort et al., 2019). On the other
hand, cyclones can strongly influence the evolution of blocking anticyclones downstream (e.g. Pfahl et al., 2015; Steinfeld
and Pfahl, 2019), which can lead to cold waves in winter and heat waves in summer (e.g. Kautz et al., 2022). Improving the
understanding and prediction of extratropical cyclone activity on subseasonal to seasonal timescales, that is, timescales of
several weeks to months, is therefore of great scientific interest and has the potential to provide more accurate forecasts of
these storms and reduce the risk of devastating events.

A range of drivers may give rise to increased prediction skill on subseasonal to seasonal timescales, including the stratosphere
(Baldwin and Dunkerton, 2001; Scaife et al., 2005; Stockdale et al., 2015) and tropical variability modes such as the El
Niño–Southern Oscillation (ENSO; Brönnimann, 2007; Scaife et al., 2014; Domeisen et al., 2015) and the Madden–Julian
oscillation (MJO; Cassou, 2008; Guo et al., 2017; Zheng et al., 2018). These drivers are often associated with external forcing
of midlatitude variability acting on longer timescales than the day-to-day weather. Such information is essential for indicating
on changes in surface weather several weeks in advance. One of these drivers that can influence storm track behavior in the
North Atlantic is the stratosphere, the layer of the Earth's atmosphere between about 10 to 50 km height.

Variability in the stratospheric polar vortex can have a long-lasting influence on surface weather (Baldwin and Dunkerton,
1999, 2001). In particular, a strengthening or weakening of the stratospheric polar vortex can lead to changes in the latitudinal
position and strength of the tropospheric jet, associated with the polarity of the NAO, for extended periods of several weeks.
Roughly two thirds of extremely weak polar vortex events, known as sudden stratospheric warmings (SSW), are followed by
a southward shift of the North Atlantic eddy-driven jet stream (e.g. Karpechko et al., 2017; Maycock et al., 2020a), generally
corresponding to a southward shift of the storm track (Baldwin and Dunkerton, 2001). For roughly one third of SSW events, the
tropospheric response is associated with a poleward shift of the tropospheric jet in the North Atlantic (Afargan-Gerstman and
Domeisen, 2020). On the other hand, a strengthening of the stratospheric polar vortex, which can result in so-called strong polar
vortex events when the stratospheric wind speed increases above a certain threshold, is generally associated with a poleward
shift of the North Atlantic storm track (Baldwin and Dunkerton, 2001; Kidston et al., 2015; Goss et al., 2021).

However, while the response of the troposphere to stratospheric forcing is generally characterized in terms of changes in the
large-scale sea level pressure pattern (Baldwin and Dunkerton, 2001), surface temperature and precipitation patterns (Butler
et al., 2017), the NAO (Charlton-Perez et al., 2018; Domeisen, 2019), atmospheric rivers (Lee et al., 2022), or shifts in the eddy-
driven jet stream (Maycock et al., 2020b), less is known about the impact of the stratosphere on the storm track on subseasonal
timescales, or how single storms might be affected. However, there are indications that anomalies in the stratospheric polar
vortex intensity can provide subseasonal prediction skill for cyclone activity in the eastern Atlantic, Northern Europe and the
Iberian Peninsula (Zheng et al., 2019). There exists a range of examples of single storms or series of storms that may have
been driven or at least made more likely by preceding stratospheric events, such as the storms following the 2018 SSW event

that triggered the persistent precipitation anomalies ending the Iberian drought (Ayarzagüena et al., 2018) or the storm series that hit the United Kingdom during the record strong Arctic Oscillation in February 2020 that was potentially linked to an extremely strong stratospheric polar vortex (Lawrence et al., 2020; Lee et al., 2020; Rupp et al., 2022). In turn, cyclogenesis can affect the downward impact from the stratosphere (González-Alemán et al., 2022). It is, however, not the goal of this study to attribute single storms to stratospheric origins. In this study, we aim to better characterize the role of the stratosphere in impacting storm tracks and extratropical cyclones.

Here, we evaluate the stratospheric influence on extratropical cyclones in a state-of-the-art Subseasonal to Seasonal (S2S) Prediction model. Cyclones are identified with a cyclone detection algorithm. Cyclone detection schemes for S2S forecasts are not yet common and their use provides a new way of evaluating forecast bias from a weather system perspective. This method is of particular interest following events that may provide windows of opportunity for extending the forecast lead time, as in the case of extreme stratospheric events.

## 2 Data and Methods

### 2.1 Reanalysis data and sub-seasonal reforecasts

In order to obtain a better understanding of how the stratosphere affects the storm tracks, we first establish the storm track response in the North Atlantic in reanalysis. We use 24-hourly instantaneous mean sea level pressure (MSLP) from the ERA5 reanalysis (Hersbach et al., 2020) to assess the cyclone frequency for the winter season (December - March) from 2000 to 2019 at a horizontal resolution of $1° \times 1°$ (~100 km). Lower-tropospheric jet intensity is identified using 24-hourly instantaneous 850-hPa zonal wind (U850) obtained from ERA5. Other atmospheric fields examined include 10-hPa zonal wind (U10) and 100-hPa geopotential height (Z100).

We compare the reanalysis results to a subseasonal prediction system, as this is the relevant tool that will be used to forecast such storms on extended-range timescales. For this purpose, subseasonal reforecasts (also called hindcasts), that is, predictions of past weather, spanning the time period from the 1st of January 2000 until the 31st of December 2019 are used from the ECMWF forecast system. The reforecasts consist of an 11-member ensemble initialized from ERA5 twice a week (on Monday and Thursday), for a period of 20 years. For example, the reforecasts of December 2, 2019 has been initialized on same date as the real-time forecast of 02/01/2020. This reforecast consists of a 11-member ensemble starting on 2nd January 2000, 2nd January 2001, ..., to 2nd January 2019 (20 years). Resolution varies with time, and is approximately 16 km up to day 15, and about 32 km after day 15. These simulations are part of the S2S Prediction research project database, an ongoing research effort for improving the forecast skill and the understanding of the climate system on subseasonal to seasonal timescales (Vitart et al., 2017).

For the major part of this study, in which the spatial characteristics of cyclone frequency are investigated, we use reforecasts from the model cycle 46R1 with 24-hourly instantaneous output. In Section 3.4, which discusses the characteristics of the full cyclone track life cycles, we use 6-hourly output from several model versions with the cycles 47R1, 47R2 and 47R3. A minimum of 6-hourly output is needed for a physically meaningful objective cyclone tracking.

The reforecasts are run for 46 days. In this study, we focus on the first four weeks of the reforecasts. For all reforecasts, week 1 is defined as 1-7 days when day 1 is the first day after initialization, week 2 is 8-14 days after initialization, week 3 is at 15-21 days, and week 4 is at 22-28 days.

## 2.2 Extratropical cyclone identification

Feature-based identification schemes have been developed for cyclones, fronts, warm conveyor belts, and jet streams. In particular, cyclone identification schemes have been widely used for reanalysis data (e.g., Sprenger et al., 2017) as well as for future projections using climate models (Harvey et al., 2020; Priestley and Catto, 2021).

Extratropical cyclones in the ECMWF model and in the reanalysis are identified from the mean sea level pressure (SLP) field using the Wernli and Schwierz (2006) detection algorithm, refined in Sprenger et al. (2017), as regions delimited by the outermost closed SLP contour enclosing one or several local SLP minima. The position of 6-hourly cyclone tracks are detected according to Sprenger et al. (2017). To neglect weak and short-lived cyclones, we only select the cyclone tracks with a lifetime of at least 24 hours and a maximum intensity (i.e., lowest sea level pressure minimum along the track) of at least 990 hPa.

For every cyclone, the application of the cyclone detection algorithm yields a two-dimensional binary field with a value of 1 at grid points that meet the cyclone criterion and 0 otherwise. Using this method, the entire area influenced by the cyclone is included within the cyclone frequency field, rather than a detection of only the cyclone core.

The climatology is then computed by temporally averaging the cyclone areas (i.e., the binary fields) (Sprenger et al., 2017). For example, a climatological value of 0.45 in DJFM indicates that this grid point is affected by a cyclone 45% of all time steps. We apply this algorithm both on the reanalysis and reforecast data. The number of the cyclone tracks can be found in Figure 6.

Cyclone frequency anomaly for each ensemble member is computed as the difference in the number of cyclones detected in the 28 days after the SSW and strong vortex events and the climatological cyclone frequency for this period. In the NH, anomalies in the tropospheric circulation after extreme stratospheric events can persist for up to 60 days after their onset (Baldwin and Dunkerton, 2001), and thus may prove to be useful for tropospheric weather and climate prediction. A period of 28 days after the onset of SSWs and strong vortex events is chosen in order to understand the initial tropospheric response and its potential for subseasonal predictions of the surface response. Composites of surface impact following stratospheric extreme events are produced by taking the ensemble mean forecast for each of the SSW events as defined below. As the reforecasts are initialized only twice per week, we examine the closest initialization date that occurs either on the same date or after each SSW, hence a date between 0 to 3 days with respect to the SSW central date. For example, for the SSW event on the 12th of February 2018 a reforecast initialized on the 13th of February is used.

## 2.3 Detection of stratospheric events

For the detection of SSW and strong polar vortex events in the reanalysis, we use daily ERA5 reanalysis for the period 2000–2019. Similar dates have been identified using ERA-Interim reanalysis (Dee et al., 2011), as found by a direct comparison of SSW dates with previous work (e.g., Butler et al., 2017).

SSWs are defined as a reversal of the zonal mean zonal winds at 60°N and 10 hPa from westerly to easterly during the extended winter period from November to March, excluding final warming events (according to the list of final warming events given in Butler and Domeisen, 2021). The central date of the SSW is defined as the first day on which the daily zonal mean zonal winds are easterly. This definition follows Charlton and Polvani (2007) and is commonly used in the literature (Butler et al., 2017). The winds must return to westerly for at least 20 consecutive days between events, to ensure that each event is counted only once. Overall, 14 SSW events are identified in our study period (2000–2019).

Strong polar vortex events are defined using a threshold of 48 m s$^{-1}$. This threshold is the 90th percentile level of the zonal wind at 10 hPa and 60°N from December through March distribution. The central date is the first day of zonal mean zonal winds above this threshold, and the winds must go below 48 m s$^{-1}$ for at least 20 consecutive days between events. Similar thresholds for the detection of strong polar vortex events can be found in the literature (e.g., Domeisen et al., 2020b; Oehrlein et al., 2020). Between 2000 and 2019, 14 strong polar vortex events are identified according to the above criterion. A full list of SSW and strong polar vortex dates that are used in this study can be found in Figure 7.

## 3 Results

### 3.1 Cyclone frequency bias in subseasonal predictions of the ECMWF model

Extratropical cyclone frequencies in the Northern Hemisphere are generally highest over the midlatitude North Pacific and North Atlantic oceans (Hoskins and Hodges, 2002, 2005; Chang et al., 2002; Sprenger et al., 2017). Over the North Atlantic, the highest cyclone frequency occurs between Greenland and Iceland, with a maximum cyclone frequency of 45% (Figure 1a).

The spatial distribution of cyclone frequency is generally well represented in the reforecasts (Figure 1b). Yet, the forecast system overestimates the cyclone frequency across midlatitudes between about 40° - 60°N (Figure 1c,d), while it underestimates the cyclone frequency along the storm track maximum and south of Greenland. The signature of the midlatitude bias in cyclone frequency is significantly larger when computed for a period of 28 days starting on the day of initialization (Figure 1d), compared to the biases over a period of 7 days (Figure 1c).

These results demonstrate the general ability of the ECMWF forecast systems to reproduce the DJFM climatological storm track, although regional biases exist, particularly in the North Atlantic and over Northern Europe, whose origin and consequences will have to be investigated further. A more in-depth analysis of subseasonal reforecast biases for Northern Hemisphere cyclone frequency and life cycle characteristics will be published in a separate future study.

### 3.2 Zonal wind response following SSW and strong polar vortex events

As a next step, we assess the prediction of the surface response following stratospheric extreme events on subseasonal timescales. We first analyze U850 following stratospheric extreme events, focusing on the differences between SSW and strong polar vortex events.

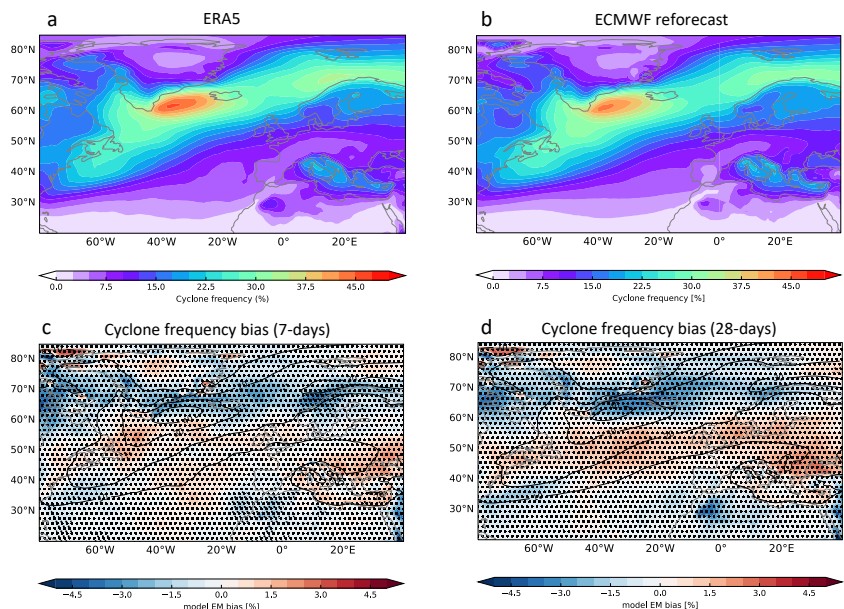

**Figure 1.** Climatology of cyclone frequency (in %) for December to March (DJFM), in (a) ERA5 reanalysis for the years 2000-2019, and in (b) ECMWF reforecasts for the same period. The climatology for the reforecasts is computed using all available initializations between the 1st of December and the 1st of March and averaged over a period of 28 days (days 1-28 with respect to the initialization date). (c) Model bias (shading, in %) according to the difference between ECMWF reforecasts and reanalysis (reforecast minus reanalysis) over a period of 7 days starting on the day of initialization, and (d) same as (c), but for a period of 28 days. Black contours in (c) and (d) show the climatological cyclone frequency in the reforecasts as shown in panel (b).

Figure 2 shows a composite of U850 after SSW and strong polar vortex events in the ERA5 reanalysis and the ECMWF reforecasts. Following SSW events, U850 anomalies in the reanalysis strengthen over the subtropical North Atlantic, particularly equatorward of 40°N, whereas a weakening of the zonal wind occurs in midlatitude, between 40° - 60°N in the North Atlantic (Figure 2a). These changes correspond to an equatorward shift of the eddy-driven jet. A similar spatial pattern of the downward impact is found in the reforecasts, however the maximum weakening occurs over a wider region in the reforecasts compared to the reanalysis, e.g., over the North Atlantic as well as over the Baltic Sea and Scandinavia (Figure 2c). Over the midlatitudes of the North Atlantic, as well as over the subtropical Atlantic, U850 anomalies are statistically significant. Note the difference in sample size between reanalysis and the reforecasts, due to the ensemble size (although ensemble members are not independent of each other).

In contrast to SSW events, U850 anomalies after strong polar vortex events show a strengthening over middle and high latitudes in the North Atlantic in the reanalysis, while a weakening of the wind occurs more equatorward, in the subtropical North Atlantic (Figure 2b). A similar pattern is observed in the reforecasts, with a significant increase of zonal wind anomalies

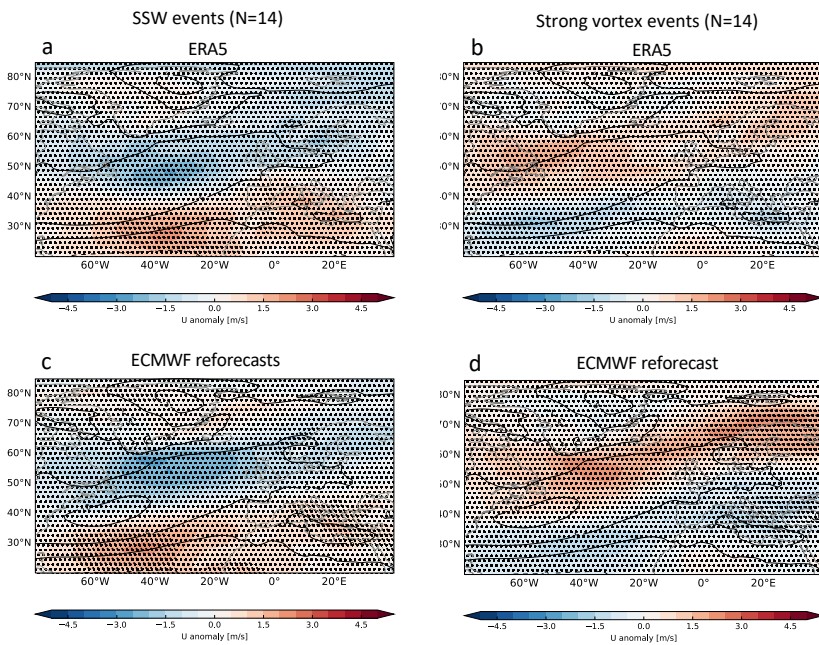

**Figure 2.** U850 anomalies (color shading; in m s$^{-1}$) following (left) sudden stratospheric warming (SSW) and (right) strong polar vortex events in (a-b) ERA5 reanalysis following stratospheric extreme events, and (c-d) ECMWF reforecasts initialized on the same date of the events or between 1 to 3 days after their first day, and averaged over a period of 28 days starting on the day of initialization. ERA5 and the reforecasts are averaged over the same period. Black contours show the climatology for DJFM. Anomalies statistically significant at the 90% confidence level based on the Student's t-test are indicated by the hatching.

over in mid- and high-latitudes compared to the reanalysis (Figure 2d). These changes coincide with a poleward jet shift in the North Atlantic region.

### 3.3 Cyclone frequency response following SSW and strong polar vortex events

After SSW events, the North Atlantic storm track in reanalysis strengthens on its southern flank relative to its climatological position and extends further into Europe (Figure 3a). This response of the North Atlantic storm track is consistent with the change in the North Atlantic jet stream, which also strengthens on its southern flank after SSWs (Figure 2a). Over Northern Europe, the cyclone frequency response in reanalysis is found to be stronger than in the reforecasts (Figure 3c), compared to reanalysis (Figure 3a).

Consistent with the zonal wind response, cyclone frequency in the strong polar vortex composite is enhanced over high latitudes in the North Atlantic (particularly, 60°-70°N) both in the reanalysis and in the model (Figure 3b,d). The maximum strengthening, however, occurs more northeastward in the reanalysis (e.g., over the Norwegian and the Barents Seas, Figure 3b) compared to the reforecasts, where most of the strengthening is between Greenland and Iceland (Figure 3d). Both the reanalysis

and the reforecasts show a significantly reduced cyclone frequency over the central North Atlantic (particularly, between 35°N

to 55°N) (Figure 3b,d).

Figure 3e,f shows the difference in cyclone frequency anomalies between reforecasts and reanalysis after SSW (Figure 3e) and strong polar vortex (Figure 3f) events. After SSWs, the model overestimates cyclone frequency over the central North Atlantic compared to the reanalysis, particularly between 40°N to 50°N and over the Norwegian Sea (Figure 3e). At higher latitudes, particularly south of Greenland, the reforecasts overestimate the reduction in cyclone frequency after SSW events compared to the reanalysis.

Overestimation of cyclone frequency anomalies in the reforecasts in comparison with reanalysis also occurs at higher latitudes (particularly between 60°N to 70°N) after strong polar vortex events (Figure 3f), with statistically significant anomalies along the tilted storm track maximum. Over the central Atlantic the reforecasts underestimate the cyclone frequency relative to the reanalysis.

The regional aspects of the cyclone frequency response after stratospheric extreme events can be more clearly characterized by analyzing the distribution of cyclone frequency anomalies over the central Atlantic after SSW and strong polar vortex events. One of the surface impacts of SSW events is the occurrence of anomalously wet conditions over western Europe and the Mediterranean and anomalously dry conditions over Scandinavia (e.g., Butler et al., 2017). These changes in precipitation patterns are likely linked to cyclone frequency over these regions. Hence, we here examine whether cyclone frequency after SSW events is indeed increased over the central and southern Atlantic region, and decreased in more poleward regions. For this purpose, we focus our analysis on the mid-latitude region (35°-55°N) of the North Atlantic (60°W-0°E). This region, located on the southern flank of the North Atlantic storm track, is where the change in cyclone frequency after SSW and strong polar vortex events is the largest (black boxes in Figure 3a,b).

Figure 4 shows the distribution of cyclone frequency anomalies following SSW and strong polar vortex events, compared to all winter days in the reforecasts (grey line). Anomalies are averaged between days 1 to 28 with respect to the central date of the stratospheric event. Over the selected region, the distribution of cyclone frequency anomalies shifts toward positive values after SSW events compared to all winter days (grey curve), both in the reanalysis (grey bins) and the reforecasts (purple) (Figure 4a). For both of these distributions, however, the shift compared to all winter days is not statistically significant at the 5%-level based on a two-sample Kolmogorov-Smirnov test between two fitted distributions.

In contrast, strong polar vortex events are followed by a shift of the distribution toward negative values of cyclone frequency anomalies, compared to all winter days (Figure 4b). This shift is significant for the reforecasts (p < 0.02) but not in the reanalysis. The statistical significance of this shift of the distribution relative to all winter days supports the important role of the stratosphere for the storm track in the North Atlantic, particularly following a strengthened polar vortex. Furthermore, the strong polar vortex composite (Figure 4b) has a narrower distribution compared to the composite of SSW events (Figure 4a), indicating a broader range of cyclone frequency anomalies following SSW events. We also note that the distribution of the reforecasts (purple bins) is slightly shifted toward more negative values compared to the reanalysis (grey bins).

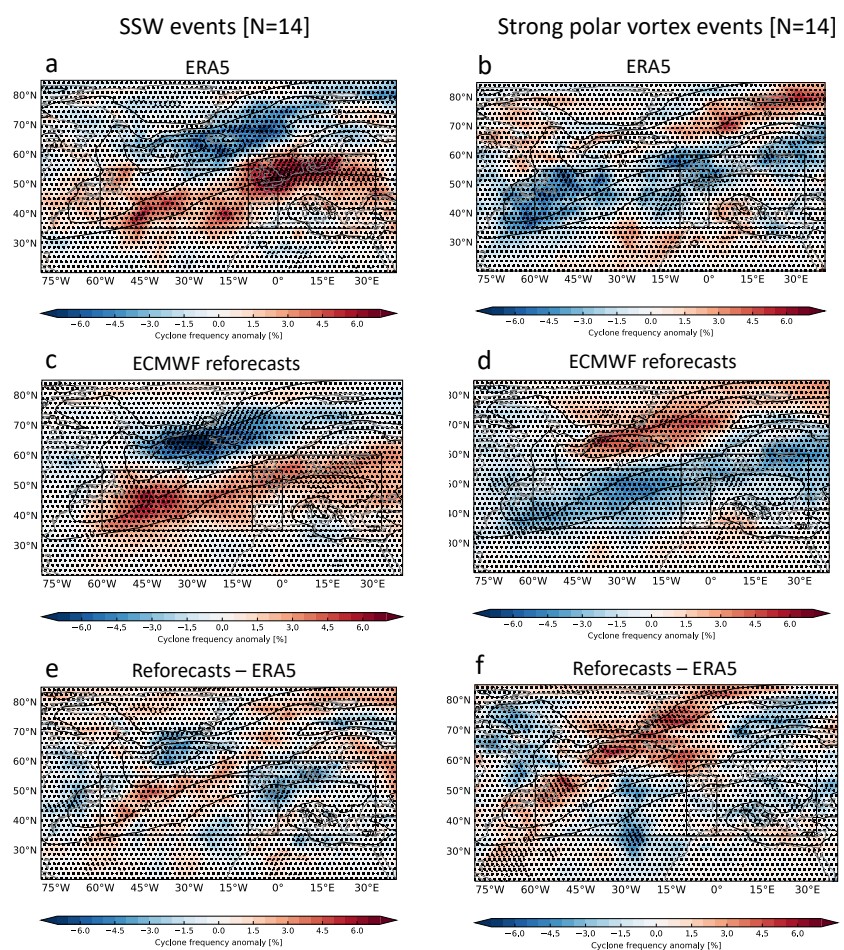

**Figure 3.** Same as Figure 2, but for cyclone frequency anomalies (in %). Reforecasts are initialized on the same date of stratospheric extreme events or between 1 to 3 days after their first day and averaged over a period of 28 days. ERA5 reanalysis is averaged over the same dates. (e-f) Differences in cyclone frequency anomalies between reforecasts and reanalysis following (e) SSW and (f) strong polar vortex events. Black contours show the climatological cyclone frequency in reforecasts for DJFM. Anomalies statistically significant at the 95% confidence level based on the Student's t-test are indicated by the hatching.

## 3.4 Cyclone life cycle characteristics following SSW and strong polar vortex events

We now investigate how the average cyclone life cycle characteristics depend on the extreme states of the stratospheric polar vortex at forecast initialization. More specifically, we analyze the spatial propagation and intensity characteristics of individual cyclone tracks, which have been identified based on an objective tracking algorithm (see methods for details). Figure 5 shows all cyclone tracks in ERA5 and in the reforecasts during the 28 days following SSW and strong polar vortex events. There are more tracks shown for the reforecasts than for reanalysis due to the use of all available ensemble members (11 members in each

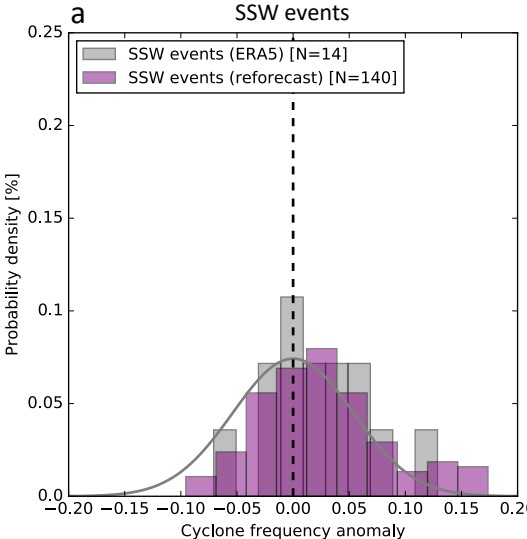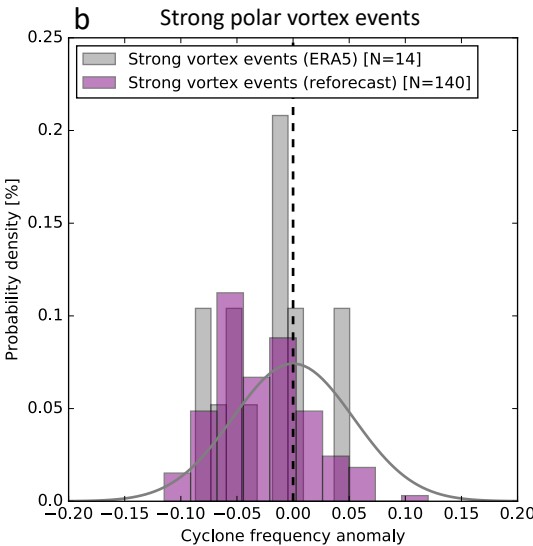

**Figure 4.** The distribution of cyclone frequency anomalies following (a) SSW and (b) strong polar vortex events in ERA5 reanalysis (grey) and in the ECMWF reforecasts (purple) in the North Atlantic (averaged over the black boxes in Figure 3). Anomalies are averaged over a period of 28 days (days 1-28 with respect to the central date of the stratospheric event). The grey curve indicates the climatological probability density for all days in DJFM in the reforecasts, represented by a normal distribution fitted using the mean and the standard deviation from the original distribution.

reforecast). Independent of the stratospheric state, the highest track densities can be found in the climatological hotspot regions along the U.S. east coast and south of Greenland (cf. black contours in Figs. 5e and 5f, which show the DJFM climatological cyclone frequency), while fewer cyclones are present over Europe and the Mediterranean. Focusing on the median track (blue and red lines, corresponding to SSW and strong polar vortex events, respectively), however, reveals a slight equatorward shift of the average cyclone propagation after SSWs, particularly over the eastern half of the North Atlantic and over Europe, which is largely in line with the findings of Baldwin and Dunkerton (2001, see their Figure 5). However, this shift is only significant (i.e., the two confidence intervals do not overlap; see caption of Figure 5 for details) in the reforecasts (Figure 5f) but not in ERA5 (Figure 5e), which might partly be related to the smaller sample size in ERA5.

We further investigate how extratropical cyclones following SSW and strong polar vortex events differ in terms of intensity as an important metric for surface impacts. The cyclones following strong polar vortex events tend to reach higher maximum intensities than the cyclones following SSW events in both ERA5, and in the reforecasts as the shift between the red (strong polar vortex) and blue (SSW) distributions in the upper left panels of Figs. 6a and 6b indicates. To determine whether these differences are significant, we split the SSW and strong polar vortex distributions into 1%-sized percentile bins, compute the difference between the percentile values of the SSW and strong polar vortex distributions for each of these bins (black line in bottom left panels of Figs. 6a and 6b), and check whether this difference is outside the corresponding 99.9% confidence

interval (grey shading; see caption of Figs. 6a and 6b for how the confidence interval is computed). According to this analysis, the difference in intensities following SSW and strong polar vortex events is highly significant in the reforecasts but not

significant in ERA5, which, however, might again be related to the smaller sample size in ERA5. To some degree, the higher intensities might be explained by the fact that the more northern cyclones following strong polar vortex events (cf. Figs. 5e and 5f) are located in regions with climatologically lower sea level pressure. Nevertheless, the cyclones following strong polar vortex events also tend to experience higher maximum intensification rates (upper right panels of Figs. 6a and 6b). The stronger intensification rates might be linked to the larger poleward component of the cyclones' propagation direction as well as the

stronger North Atlantic jet following strong polar vortex events (cf. Figure 2), which both correlate with cyclone intensification (e.g., Rivière et al., 2012; Tamarin and Kaspi, 2016; Besson et al., 2021). However, the differences in maximum intensification between SSW and strong polar vortex events are not significant in ERA5 and only significant for the most strongly intensifying cyclones (i.e., the lower percentiles) in the reforecasts (bottom right panels of Figs. 6a and 6b).

### 3.5 Reforecast performance for regional cyclone frequency after SSW and strong polar vortex events

Next, the ability of the subseasonal ensemble reforecasts in predicting North Atlantic cyclone frequency after SSW or strong polar vortex events is examined (Figure 7). We focus on two sectors: the central region of the North Atlantic (60°W-0°E, 35°-55°N, black box in Figure 3) and Europe (10°W-33°E, 35°-60°N), where anomalous cyclone frequencies are expected following SSW and strong polar vortex events (cf. Figure 3). Red bars in Figure 7a indicate the proportion of ensemble members that show an average increase in cyclone frequency over this region, whereas blue bars indicate a decrease. For simplicity, 10

ensemble members (i.e., 10 perturbed simulations of the forecast system, excluding the control run) are analyzed for each event.

#### 3.5.1 North Atlantic

The majority of SSW events are followed by an enhancement of cyclone frequency in the central North Atlantic in the reanalysis (10 out of 14 events) as indicated by the red stars in Figure 7a. The cyclone frequency response following these events is

generally well predicted, with an increase of cyclone frequency predicted by more than 60% of the ensemble members in the reforecasts (Figure 7a). In contrast, the response after SSW events with a decrease in cyclone frequency over the central Atlantic tends to be less predictable, with the majority of ensemble members predicting a decrease in only 1 out of 4 SSW events (Figs. 7a).

Strong polar vortex events, on the other hand, tend to be followed by a decrease in cyclone frequency in the reanalysis (10

out of 14 events, indicated by the blue stars in Figure 7c). This response is generally well captured by the reforecasts, with 60% or more of the ensemble members predicting a reduction in cyclone frequency after strong vortex events (Figure 7c).

On average over all events, about 60% of ensemble members predict a positive sign of the cyclone frequency anomaly in the central Atlantic after SSW events, compared to 40% of ensemble members predicting a negative anomaly. The opposite ratio between ensemble members with an enhanced versus reduced cyclone frequency response is found after strong polar

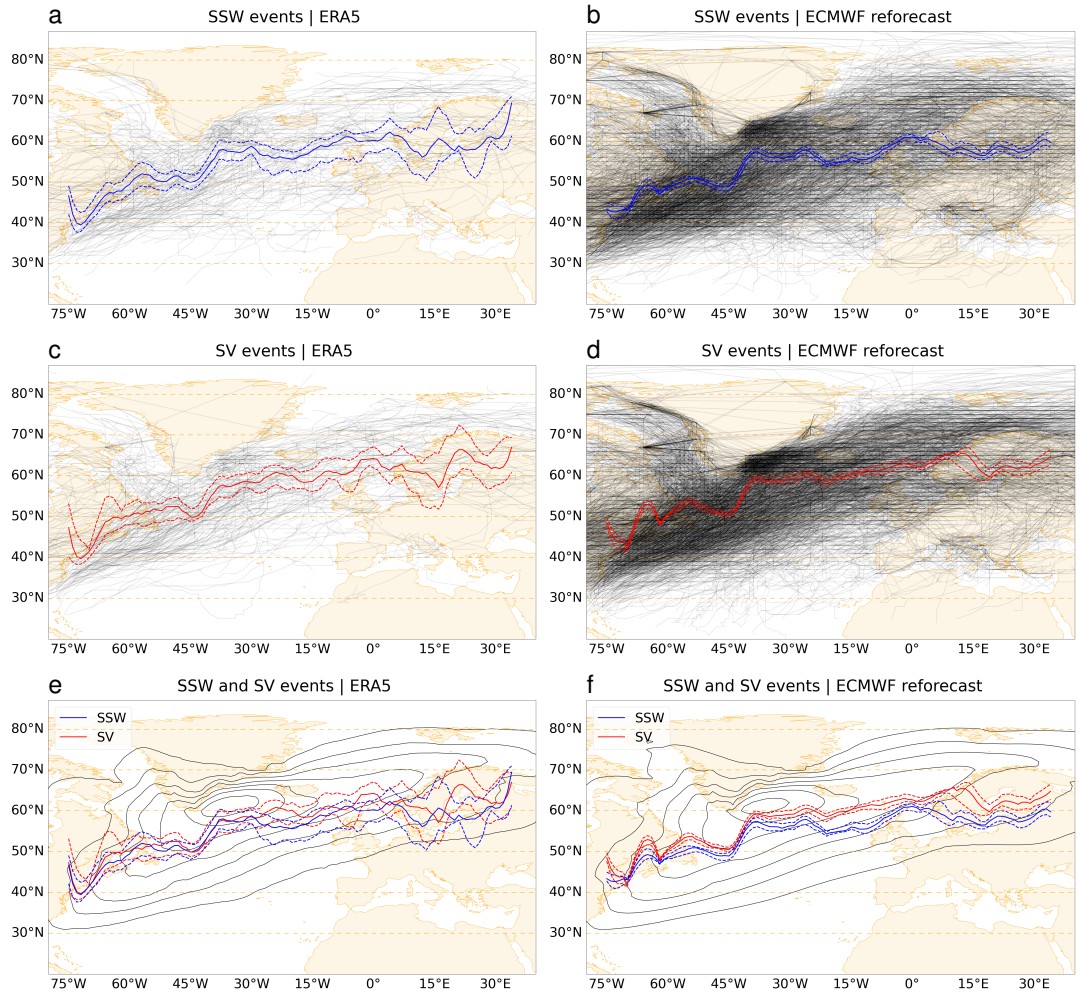

**Figure 5.** Statistics of individual cyclone tracks with a lifetime of at least 24 hours and a maximum intensity of at least 990 hPa reached within the North Atlantic-European domain in ERA5 (a, c, e) and in the reforecasts (b, d, f). The individual tracks occurring within 28 days after the SSW and strong polar vortex (SVs) events are shown in black (a - d) and the corresponding median latitude (solid) of all tracks in 1-degree longitudinal bands and its 90% confidence interval (dashed) are shown in blue and red. The confidence interval is obtained from a bootstrapped distribution of median latitudes (based on 1000 random resamples of the tracks with replacement). The DJFM cyclone frequency climatology is shown as black contours in (e) and (f).

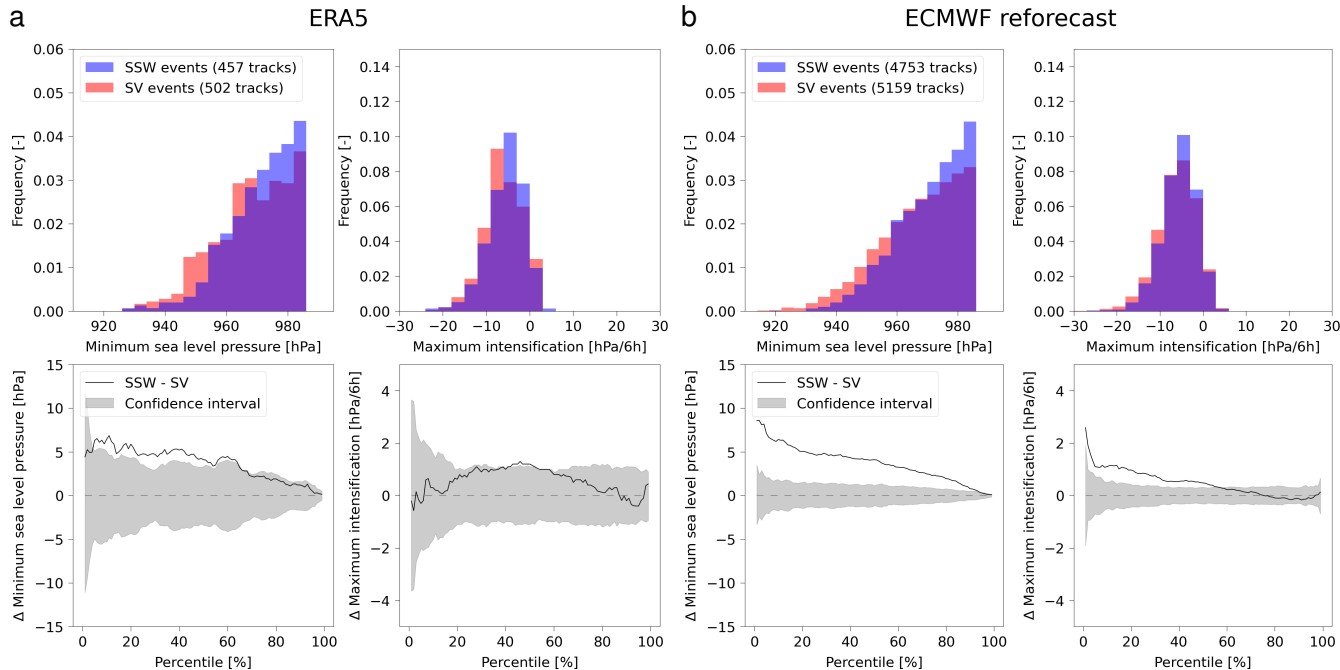

**Figure 6.** Frequency histograms for maximum intensity (defined as the lowest sea level pressure minimum along the track) and maximum 6-hourly intensification along the track of the cyclones occurring within 28 days after the SSW (blue) and strong polar vortex events (red) in ERA5 and the reforecasts are shown in the first row. The corresponding differences between the percentile values of the SSW and strong polar vortex distributions are shown by the black lines in the second row (see text for details), complemented by their 99.9% confidence intervals in grey. The confidence intervals are obtained as follows: all data points of both the reforecasts and ERA5 are combined into one distribution and this distribution is randomly shuffled. The shuffled distribution is then split into two new equally sized distributions mimicking the "ERA5" and "reforecast" distributions, and the percentile-wise difference between these two random distributions is computed in the same way as for the original distribution. This procedure is repeated 10000 times to obtain a distribution of differences for each 1%-sized percentile bin.

vortex events. For SSWs, this ratio corresponds to the percentage of SSW events with a canonical downward response, i.e., an equatorward shift of the North Atlantic jet (e.g., Afargan-Gerstman and Domeisen, 2020).

       Another way to evaluate the model performance in predicting anomalies of cyclone frequency is by computing the percentage of hits for SSW and strong polar vortex events (Figs. 7b,d). A hit is defined when more than 50% of the ensemble members predict the correct sign (i.e., the same as in reanalysis) of the cyclone frequency anomaly over the selected region.

The ensemble-mean prediction shows that the majority of SSW events with an enhanced cyclone frequency response in the midlatitude Atlantic are well predicted (90% of SSWs) in terms of the sign of their downward impact, compared to only 25% of SSW events with a reduced cyclone frequency response (Fig 7b). For comparison, strong polar vortex events tend to have higher success rates than SSWs, with more than 75% of strong polar vortex events having a successfully predicted cyclone frequency response (Figure 7d). These success rates are found for strong polar vortex both with an enhanced or reduced response.

### 3.5.2 Europe

Similar to the North Atlantic, we find that the majority of SSW events are followed by an enhancement of cyclone frequency over Europe in the reanalysis (12 out of 14 events, Figure 8a), whereas strong polar vortex events are generally followed by a decrease in cyclone frequency over Europe (8 out of 14 events, Figure 8b). However, the number of strong vortex events with a reduced cyclone frequency response is lower over Europe compared to the North Atlantic (8 versus 10 events). In terms of the percentage of hits, SSW events with an enhanced cyclone frequency response over Europe are are found to be well predicted (80% of SSWs), compared to only 50% of SSW events with a reduced cyclone frequency response (Fig 8b). This ratio is higher over Europe compared to the North Atlantic (Fig 7b), where only 25% of SSW events with a reduced cyclone frequency response are successfully predicted (however, the number of events with such response is larger).

Strong polar vortex events, on the other hand, exhibit a high number of hits compared to SSWs over the European region, with more than 90% of strong polar vortex events having a successfully predicted reduced cyclone frequency response (Figure 8d). Success rates, however, are lower over Europe compared to the North Atlantic for strong polar vortex events with enhanced cyclone frequency response (30% of strong vortex events over Europe, compared to 50% over the North Atlantic). Overall, these differences in predictability over Europe compared to the North Atlantic suggests that SSWs are characterized by higher success rates over Europe (for both enhanced and reduced cyclone response).

### 3.6 Evaluation of cyclone frequency prediction on weekly timescales

Next, in order to better understand the time evolution of the cyclone frequency response to stratospheric influences, we evaluate the hits for each week separately, starting from the central date of the SSW or strong polar vortex event (Figure 9). For the majority of SSW events, the percentage of hits is lower in weeks 3-4 compared to weeks 1-2 (Figure 9a). Out of 14 SSW events, several events have a low hit rate even in week-1 (e.g., 20 March 2000, 30 December 2001, 24 March 2010). strong polar vortex events, on the other hand, are followed by a high hit rate for week-1, with a 100% hit rate for all strong polar vortex events except one (11 February 2005; Figure 9b). The hit rate rapidly drops in the subsequent weeks.

These differences between SSW and strong polar vortex events again suggest that the model encounters more difficulties in predicting the cyclone frequency response after SSW events compared to strong polar vortex events. The reasons for this behavior can vary between the events: For example, the SSW event of 22 February 2008 was followed by a reduction in cyclone frequency over the central Atlantic (as indicated by the red star in Figure 7a); while the reforecasts fail to predict the sign of the cyclone frequency response as averaged over a 28-day period after the SSW central date (Figure 7a), the forecast model prediction for weeks 1 and 2 is in good agreement with observations (Figure 9a). However, none of ensemble members predicted a positive cyclone frequency response in week-3, and the hit rate remained relatively low in the following week, suggesting that the majority of ensemble members predicted a weakening of the cyclone frequency in weeks 1 and 2 following the SSW.

Overall, this analysis shows that while 70% of the reforecasts capture the sign of the cyclone frequency response over the North Atlantic during weeks 1-2 after SSWs, less than 50% of the reforecasts capture the response during weeks 3-4. The

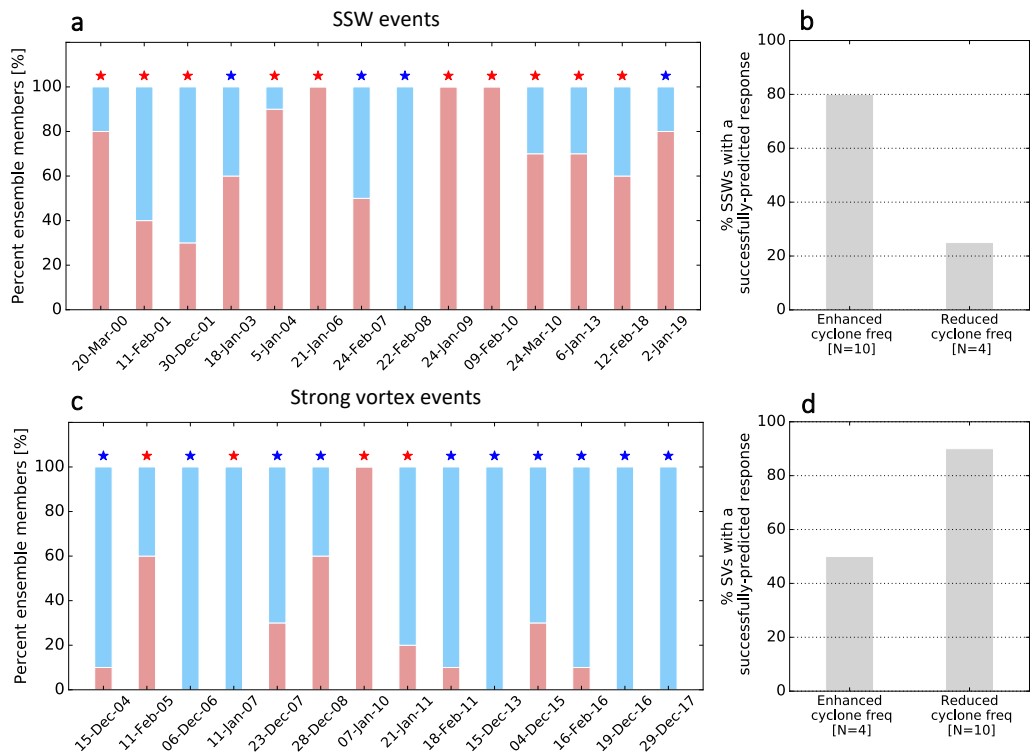

**Figure 7.** (a,c) Bars represent the percentage of ensemble members that predict an enhancement (red) or a reduction (blue) of cyclone frequency anomaly over the central North Atlantic (60°W-0°E, 35°-55°N, black box in Figure 3a) after (a) SSW and (c) strong polar vortex events in the ECMWF reforecasts. The x-axis in (a,c) indicates the central dates of the stratospheric events. Anomalies are averaged over days 1-28 of the reforecast. Red and blue asterisks indicate the average response based on ERA5, with red (blue) indicating an increase (decrease) of cyclone frequency anomaly over this region. (b,d) The percentage of events where more than 50% of the ensemble members predict the correct sign of the cyclone frequency anomaly over the midlatitude North Atlantic region, for (b) SSW, and (d) strong polar vortex events.

cyclone forecasts following strong polar vortex events are generally more successful, with more than 90% of the reforecasts predicting the response during week 1, and around 60% capturing the response in the following weeks.

## 3.7 Dynamical aspects of successful and unsuccessful predictions

Here, the relationship between ensemble members predicting the observed cyclone frequency response after SSW and strong polar vortex events and the large-scale atmospheric circulation patterns at the surface and in the lower stratosphere is examined. We use mean sea level pressure (MSLP) and geopotential height anomalies at 100 hPa (Z'100) in the aftermath of the stratospheric events to evaluate the predictions.

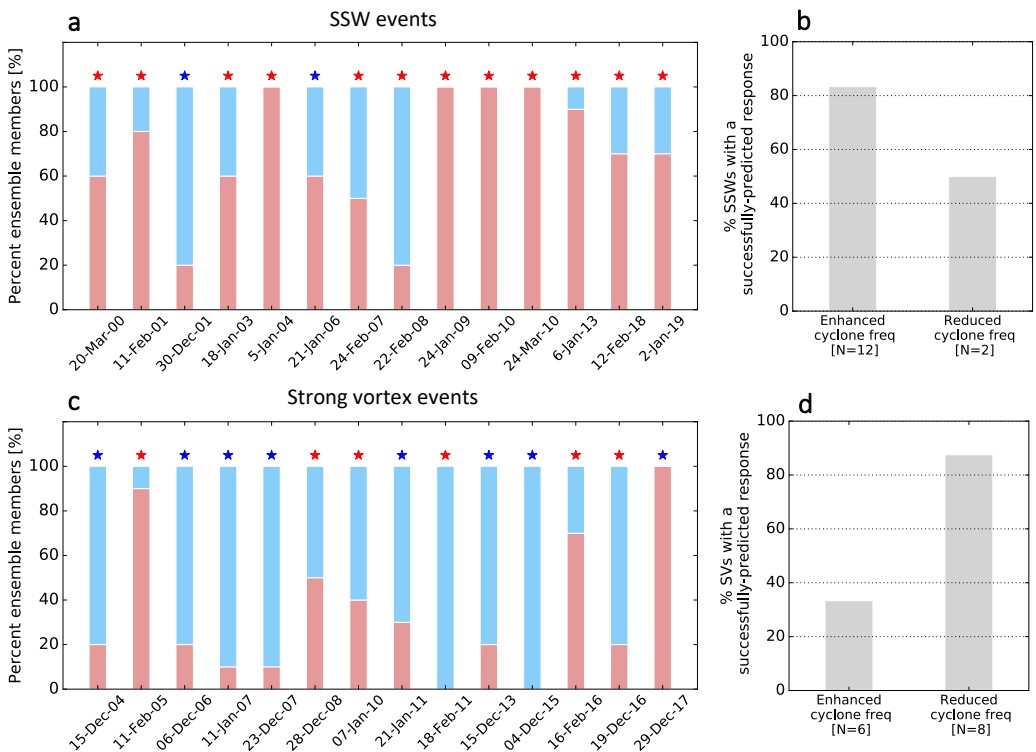

**Figure 8.** Same as Figure 7, but for Europe (10°W-33°E, 35°-60°N, black box in Figure 3a).

Figure 10 shows the time evolution of the cyclone frequency prediction averaged over the North Atlantic (60°W-0°E) after SSW and strong vortex events (Figure 10a,b, respectively). Only events with a canonical downward response (according to the reanalysis) are used: SSW events with an enhanced cyclone frequency in the midlatitude North Atlantic, and strong polar vortex events with a reduced cyclone frequency in the same region.

For each reforecast, the ensemble members are separated into two subgroups according to the success of their prediction. A successful prediction (indicated by the blue curves in Figure 10a,b) is defined here per ensemble member that predicts the observed sign of the cyclone frequency anomaly in the North Atlantic (based on a 28-day average of the response after the onset of SSW or strong vortex events, respectively). In contrast, unsuccessful predictions (indicated by the orange curves in Figure 10a,b) are defined as members that do not predict the observed sign on the response for the same period.

We find that out of 100 ensemble members of SSW events with a canonical surface response (i.e., enhanced cyclone frequency in the midlatitude North Atlantic), 74% successfully predict the sign of the downward response, whereas 26% are unsuccessful in predicting the correct sign. For strong polar vortex events with a canonical surface response (i.e., reduced cyclone frequency in the midlatitude North Atlantic), 85% out of 100 ensemble members result in a successful prediction, and 15% in an unsuccessful prediction. Furthermore, we find that cyclone frequency anomalies in unsuccessful predictions diverge from the

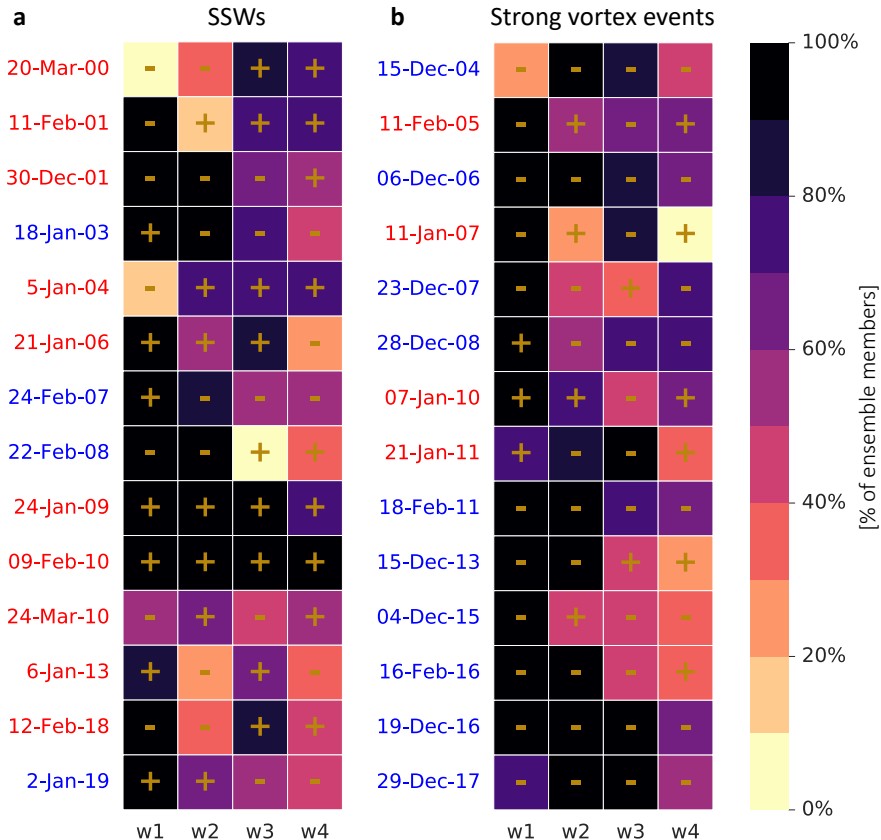

**Figure 9.** Percentage of ensemble members that predict the observed cyclone frequency response over the central North Atlantic (60°W-0°E, 35°-55°N) after (a) SSW and (b) strong polar vortex events in the ECMWF reforecasts. Anomalies are averaged for every week in the reforecast (w1 is between days 1-7, w2 between days 8-14, etc) with respect to the central date of the event. For each week, the observed response is indicated by a "+" ("-") sign corresponding to an increase (decrease) of cyclone frequency anomaly in the selected region. A red (blue) date corresponds to an average increase (decrease) of cyclone frequency anomaly during weeks 1-4.

successful forecasts within the first 2-4 days with respect to the event central date.

Anomalies of MSLP for successful and unsuccessful predictions of the cyclone frequency response in the North Atlantic after SSW and strong vortex events are shown in Figure 10c,d and Figure 10e,f, respectively. As expected, SSW events with a successful canonical response are characterized by a dipole pattern of MSLP that projects onto a negative NAO pattern (Figure 10c). The unsuccessful predictions, however, are characterized by a high pressure pattern dominating the North Atlantic sector (Figure 10d).

After strong polar vortex events, MSLP anomalies project onto a positive NAO pattern following a successful prediction of the

cyclone frequency response (Figure 10e), whereas the dipole pattern is shifted further southeast for an unsuccessful prediction of the surface response (Figure 10f).

In contrast to the MSLP anomaly pattern, the Z'100 pattern shows positive (negative) anomalies over the pole for both successful and unsuccessful prediction of the surface response after SSW (strong polar vortex) events (Figure 10g,h and Figure 10i,j, respectively). For SSW events, positive polar cap anomalies of Z'100 are found to be stronger for SSWs with a successful prediction, compared to the unsuccessful predictions, consistent with previous studies on the importance of lower stratospheric geopotential height anomalies for the downward impact (e.g., Karpechko et al., 2017; Afargan-Gerstman et al., 2022). For the strong polar vortex events, negative geopotential height anomalies over the polar cap are found to have a more zonally symmetric pattern at 100 hPa in the case of a successful prediction (Figure 10i), and a more asymmetric pattern for unsuccessful predictions (Figure 10j).

Thus, we find that ensemble members with a successful prediction of the canonical downward influence in the Atlantic differ from unsuccessful members mostly in their representation of tropospheric circulation anomalies after SSW events, indicating that the troposphere plays a dominant role in a successful prediction of the downward impact of stratospheric anomalies after SSW events, as e.g. indicated by Domeisen et al. (2020c). Following strong polar vortex events, however, members with successful predictions differ from unsuccessful members in both their tropospheric and lower stratospheric anomalies.

To further understand the difference in tropospheric circulation between successful and unsuccessful predictions of the cyclone frequency response, we analyze the time evolution of MSLP and zonal wind anomalies for successful and unsuccessful predictions after SSW and strong vortex events (Figures 11 and **??**). Anomalies are plotted for every week in the reforecast with respect to the central date of the event.

Successful predictions of the canonical downward impact after SSW events is characterized by a dominant dipole pattern of MSLP anomalies over the North Atlantic (with a high pressure anomaly over Greenland, and low pressure anomaly over the midlatitude Atlantic), persisting between week 2 to week 4 (Figure 11a). In contrast, unsuccessful predictions lack the occurrence of a persistent dipole pattern (Figure 11b), and are characterized by a high pressure anomaly over Northern Europe (in week 1) and the North Atlantic (in weeks 2 and 3).

In comparison, successful predictions of the canonical downward impact after strong polar vortex events are followed by a similar but opposite dipole pattern of MSLP anomalies in the North Atlantic (Figure 11c). However, this dipole pattern persists only for weeks 1 and 2 (with weaker reappearance in week 4). Unsuccessful predictions, for comparison, exhibit larger natural variability of the surface circulation, with a different anomalous MSLP pattern in every week of the reforecast (Figure 11d).

Similarly, successful predictions of the canonical downward impact after SSW events are found to b associated with a persistent equatorward shift of the North Atlantic jet in weeks 1 to 4, as shown by the zonal wind anomalies at 850 hPa (Figure 12a), while unsuccessful predictions show a persistent pattern only in weeks 2 and 3 (Figure 12b). In contrast to SSW event, successful and unsuccessful predictions of the canonical impact after strong vortex events both exhibit a persistent response between week 1 and week 4 (Figure 12c,d).

Figure 13 shows the time evolution of the ensemble mean prediction for cyclone frequency anomaly (Figure 13a,b) averaged over the North Atlantic (60°W-0°E) for SSW and strong vortex events, respectively. All reforecasts are initialized after the onset of the events (see Methods section for details). The ensemble mean is computed for each event separately, and then averaged over all selected events.

The ensemble mean shows the enhancement of cyclone frequency in the midlatitudes after SSW events (solid contours in
Figure 13a). After initialization, cyclone frequency is increased between 45°N to 60°N. Starting from day 5, positive anomalies are observed further equatorward (mostly between 30°N to 55°N), consistent with an equatorward shift of the storm track. On the other hand, ensemble predictions after strong vortex events show a decrease in cyclone frequency in the midlatitude region (30°N to 55°N), starting at day 0 (dashed contours in Figure 13b), indicative of a average poleward shift of the storm track in this region.

Next, we examine the ensemble spread for these reforecasts. The ensemble spread is represented by the standard deviation with respect to the ensemble mean. As for the ensemble mean, the ensemble spread shown in Figure 13 is averaged over all events with a canonical downward response. Reforecasts after SSWs exhibit a relatively small spread in the first days after the onset of the SSW events, however the spread increases gradually with time, in particular after day 10 (Figure 13a). An additional increase in ensemble spread occurs after day 20. Throughout its evolution, the spread is largest between 45°N and 60°N,
which marks the transition zone between positive and negative cyclone frequency anomalies after SSW events. Interestingly, the ensemble spread after strong vortex events is largest at high latitudes, between 55°N and 70°N (Figure 13b), which is the region corresponding to the poleward shift of the ensemble mean.

Overall, the largest spread is found between 50°N and 65°N for SSW events, and between 60°N and 65°N for strong vortex events. While SSW and strong vortex events generally exhibit similar but opposite tropospheric response, differences in the
predictability of their response can be found, as shown by the ensemble spread beyond 10 days.

## 4    Conclusions

Our results show that stratospheric extremes can have a clear impact on the storm track and on cyclone occurrence and tracks, with clear differences between weak and strong stratospheric polar vortex events. In more detail, our results can be summarized as follows:

– The model shows the expected response of the North Atlantic jet stream following stratospheric extreme events (i.e., an equatorward shift after SSW events and a poleward shift after strong polar vortex events) when averaging over all events.

– The North Atlantic storm track (measured by the local frequency of cyclone occurrence) exhibits a behavior consistent with the jet, i.e., an enhanced (reduced) cyclone frequency equatorward of the climatological storm track maximum after SSW (strong polar vortex) events.

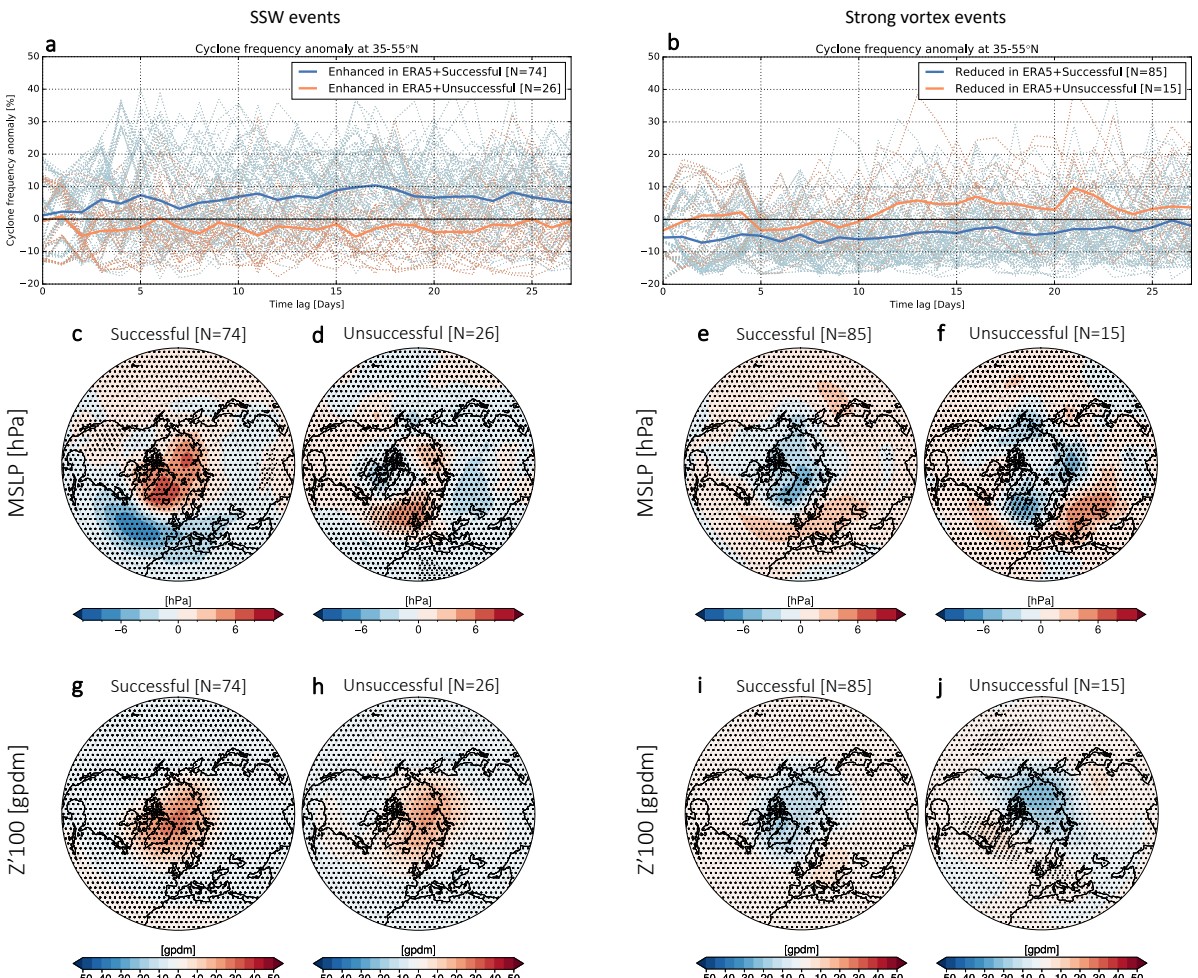

**Figure 10.** (a,b) Time evolution of cyclone frequency anomaly (in %), zonally averaged over the midlatitude North Atlantic, following (a) SSW events, and (b) strong polar vortex events with a canonical surface response (see text for definition). Ensemble members with a successful (blue) and unsuccessful (orange) prediction of cyclone frequency are highlighted. The bold line is the ensemble mean of each composite. The numbers in the brackets of the legend show the number of events in each composite. (c-f) Composites of MSLP anomalies (in hPa) for (c,e) successful and (d,f) unsuccessful prediction after SSW and strong polar vortex events, respectively. (g-j) Same as (c-f), but for geopotential height anomalies of the 100 hPa surface (Z'100; in gpdm). Anomalies statistically significant at the 90% confidence level based on the Student's t-test are indicated by the stippling.

– The strongest biases in the cyclone frequency model response are observed over northwestern Europe after SSW events, where cyclone frequency is underestimated, and after strong polar vortex events to the south and east of Greenland, where cyclone frequency is overestimated.

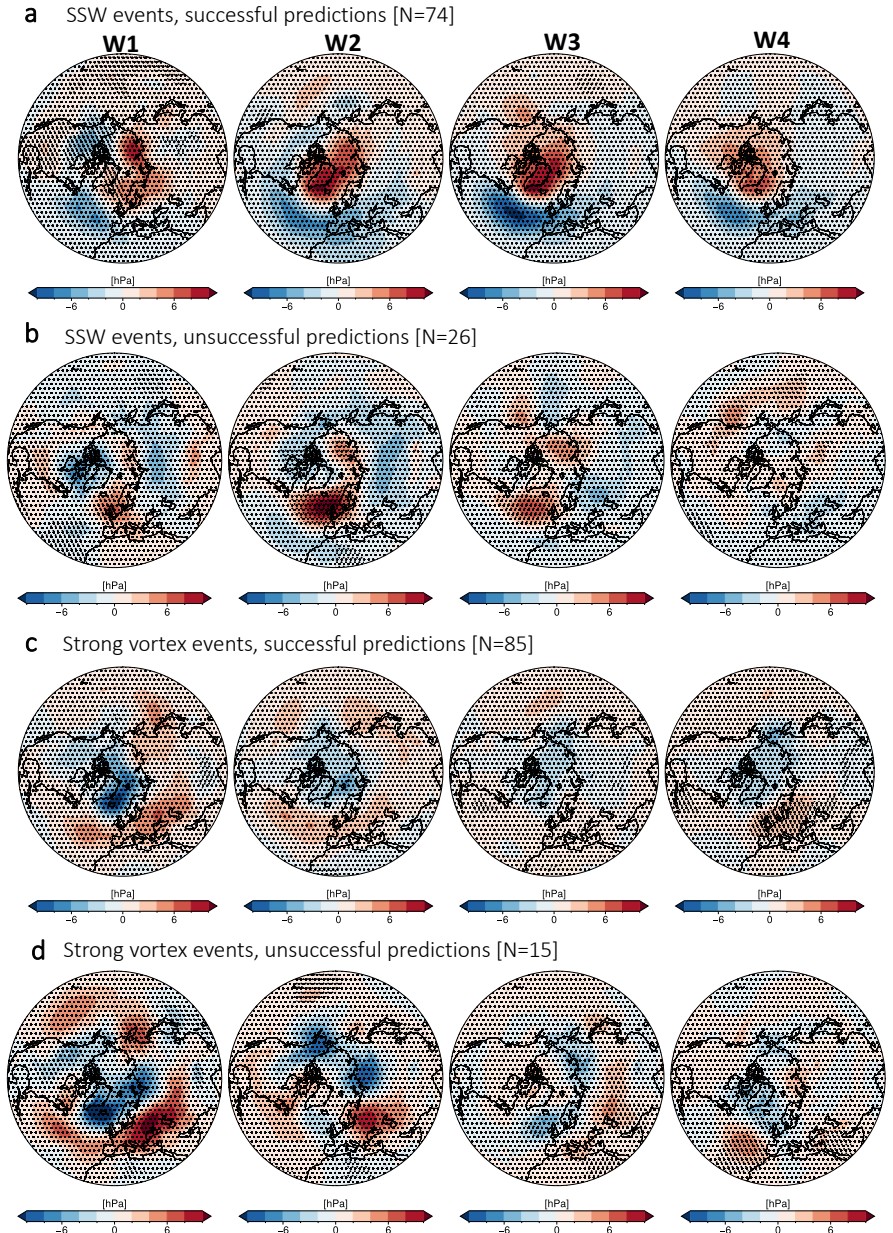

**Figure 11.** Same as panels (c-j) in Figure 10, but for MSLP anomalies in every week in the reforecast (w1 is between days 1-7, w2 between days 8-14, etc) with respect to the central date of the event. Anomalies are shown for (a,c) successful and (b,d) unsuccessful predictions after SSW and strong polar vortex events, respectively.

– The southward shift after SSWs compared to strong polar vortex events also manifests itself over the eastern North Atlantic when defining the storm track by the median of individual cyclone tracks. Furthermore, the cyclones after

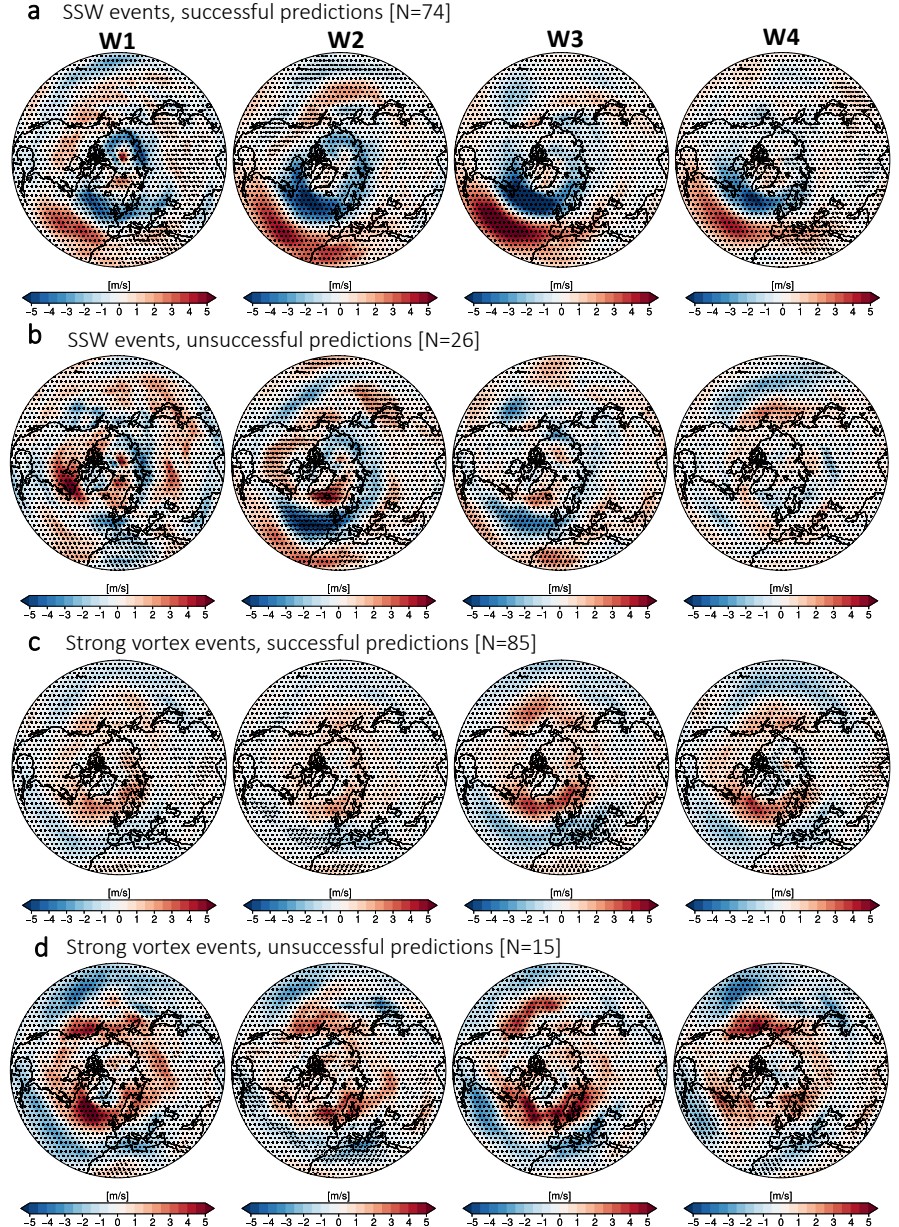

**Figure 12.** Same as Figure 11, but for 850-hPa zonal wind anomalies for (a,c) successful and (b,d) unsuccessful predictions after SSW and strong polar vortex events, respectively.

strong polar vortex events intensify more strongly and reach higher intensities than after SSW events. However, both the differences in cyclone track location and cyclone intensity are only significant in the reforecasts but not in the reanalysis (with the exception of the significantly stronger cyclone intensities following strong polar vortex events also

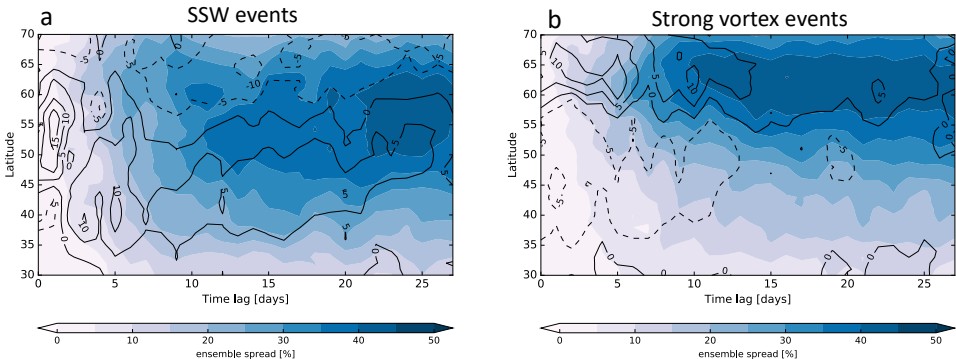

**Figure 13.** Ensemble mean prediction (black contours; negative values are dashed) and ensemble spread (color shading) for zonal mean cyclone frequency anomaly (in %), averaged over the North Atlantic (60°W-0°E) for reforecasts initialized after (a) SSW and (b) strong polar vortex events. Only events with a canonical surface response in the reanalysis are included in the composites.

in reanalysis). A larger sample size would be required to determine whether this result is simply due to the smaller sample size in ERA5 or whether this might indicate a slight overconfidence of the reforecasts in predicting the storm track response.

- For individual events, the sign of a canonical (expected) response, i.e., an enhancement (reduction) in cyclone frequency after SSWs (strong polar vortex events) in the central North Atlantic is generally well predicted (above 80% of all events).

- For SSW and strong polar vortex events without a canonical response, the enhanced cyclone frequency in the midlatitude North Atlantic is well predicted in 50% of all strong polar vortex events, while the reduced response is predicted only in 25% of all SSW events.

- SSWs exhibit significantly more variability between events with respect to predictability. In particular, the surface response to strong polar vortex events can almost always be predicted in the first lead week, with a decrease in predictability thereafter, while the predictability behavior for SSW events is much less uniform between events.

- A successful prediction of the canonical response depends more strongly on a correct representation of the state of the troposphere than the lower stratosphere at the time of the SSW event, while for strong vortex events both the lower stratosphere and the surface state are important.

Concluding, the model successfully represents the surface cyclone frequency response after most strong polar vortex events, especially for short lead times. For SSW events however the results are more mixed: The model is generally more successful in predicting the cyclone frequency after SSWs when the response to the stratospheric events exhibits the canonical response, i.e. an equatorward shift of the storm track. This result points towards a possible overconfidence of the model with respect to reanalysis to predict the canonical response after SSW events, which is however only warranted for about two thirds of SSW

events. This is consistent with previous findings on the prediction of the NAO following stratospheric events, which tends to over-predict the occurrence of the negative NAO phase after SSW events (Kolstad et al., 2020, 2022), leading to a poor prediction of surface temperatures over Europe after SSW events in these cases(Domeisen et al., 2020a).

This relation between cyclone activity and variations in the stratospheric polar vortex is consistent with previous studies on the subseasonal prediction of wintertime extratropical cyclones, particularly over the eastern Atlantic, Europe and East Asia (Zheng et al., 2019). We find that the majority of ensemble members well predicted the cyclone frequency over the midlatitude Atlantic and Europe in the period that followed stratospheric extreme events, i.e., strengthening of the cyclone frequency after SSW events, and the opposite response after strong polar vortex events. While the tropospheric response following these two

types of stratospheric events is overall similar but of opposite signs, we also find differences in their downward impact. For example, the downward influence after SSW events exhibits larger uncertainty in midlatitudes than the corresponding influence of strong polar vortex events. These results are in agreement with Rupp et al. (2022) that found the downward influence of positive stratospheric zonal circulation anomalies to be less robust than negative anomalies, as well as asymmetries in the stratosphere-troposphere wave coupling during these events.

Further investigation of the role of the stratosphere in subseasonal storm track and cyclone variability will have significant benefits for improving the prediction of extratropical cyclones and large-scale weather patterns in these regions. Understanding of the links between extratropical cyclones and persistent atmospheric circulation patterns, as forced by the downward impact of the stratosphere, has the potential to provide more accurate forecasts of intense storm impacts, and helps to reduce the risk against damages incurred by such extreme events.

*Code and data availability.*  The ECMWF reforecast data have been obtained from the ECMWF server at https://apps.ecmwf.int/datasets/data/s2s. The ERA5 reanalysis has been obtained from the ECMWF server (https://www.ecmwf.int/en/forecasts/datasets/reanalysis-datasets/era5, last access: June 2022). ERA-Interim reanalysis has been obtained the ECMWF server (https://apps.ecmwf.int/datasets/data/interim-full-daily/levtype=sfc, last access: June 2022). Cyclone frequency data are available from the corresponding author upon request.

*Author contributions.*  H.A.-G. performed the analysis of the cyclone frequency in the S2S reforecasts, and writing of the manuscript. D.B.
performed the analysis of the cyclone life cycle characteristics, and contributed to the writing of the manuscript. C.O.W. provided the S2S reforecasts for the cyclone frequency analysis and contributed to the interpretation of the results. M.S. applied the cyclone detection scheme for the S2S reforecasts and contributed to the interpretation of the results. D.I.V. contributed to the analysis and interpretation of the results, and to writing of the manuscript. All authors contributed to editing of the final manuscript.

*Competing interests.*  The authors declare that they have no conflict of interest.

*Acknowledgements.* H.A.-G. acknowledges funding from the European Union's Horizon 2020 research and innovation programme under the Marie Sklodowska-Curie (MSC) (grant agreement No. 891514). Support from the Swiss National Science Foundation through project PP00P2_198896 to D.D. is gratefully acknowledged. D.B. acknowledges funding from the Swiss National Science Foundation (grant no. 205419). C.O.W. acknowledges funding by the Research Council of Norway through the Climate Futures center (Grant 309562). We further want to thank two anonymous referees and the editor Irina Rudeva for their very constructive and helpful comments on earlier versions of

this manuscript. We further thank Rachel Wu for her help with obtaining the ECMWF reforecasts and for useful discussions.

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
