# Peer review of "Stratospheric influence on the winter North Atlantic storm track in subseasonal reforecasts"

_Weather and Climate Dynamics, 2022_

## Referee Comment (RC1)

**Review: Stratospheric influence on the winter North Atlantic storm track in subseasonal reforecasts**

Hilla Afargan-Gerstman, Dominik Büeler, C. Ole Wulff, Michael Sprenger, and Daniela I.V. Domeisen

December 19, 2022

**Summary**

This analysis identified and examined the impact of stratospheric extremes on the extratropical storm track, cyclone frequency and cyclone life cycle characteristics at subseasonal time range. The results show that events with a canonical response after weak and strong stratospheric polar conditions are better predicted compared to the events without the canonical response. This possibly leads to over-prediction by the model as it tends to forecast the expected response. The predictability after SSW events is less uniform across events compared to the predictability of the surface response after SPV states. In my opinion, while the response analysis may be not very novel, it provides solid quantification. The most interesting part discusses the predictability of the response and provides new insights into the driving mechanisms. Therefore, I recommend this manuscript for publication, but I also suggest a few comments below.

**Major comments**

- Had you considered to split the 28 days period into, for example, two parts (week 1-2 and week 3-4) in the first part of the manuscript? It would probably be interesting to look at least at U850 and cyclone frequency anomalies especially having in mind the results of the predictability part.
- The box position choice does not seem well explained. You say that in this region the increase in cyclone frequency is biggest after SSW events (L185), but the anomalies are biggest only in reforecasts (Fig. 3a). Moreover, the anomalies are biggest and statistically significant over the Northern Europe in reanalysis (Fig. 3c), which can be also seen in reforecasts. Had you considered taking a box more to the north-east of its current position? Also, was your choice of the box position based only on the anomalies after the SSW events? I see that the biggest anomalies after the SPV cases are still concentrated inside the box (Fig. 3b), but maybe this can be pointed out in the text.

**Minor comments**

L32 "predication" -> prediction

L34 Add brackets to citation

L48-51 This sentence seems to repeat the information given above, please consider removing it or rephrasing the repetition

L88 As reforecasts are initialized in conjunction with real-time forecasts, could you provide here the dates/years of the real-time forecasts? You indicate below the model versions used, but this is potentially confusing, as, for example, the 46R1 version does not have reforecasts for December 2019

L97 Please consider adding "… in the ECMWF model and in reanalysis…" if you used the same algorithm

L100 Consider adding here a remark that the number of the cyclone tracks can be found in Fig.6

L104 DJF -> DJFM for consistency throughout the text

L108 Did you use cross validation when computing the anomalies for each ensemble member?

L108 While I understand the choice of 28 days, it could be better clarified here for better understanding

L117 I wonder of you checked if there is no difference indeed when using ERA-Interim or ERA-5?

L120 Please specify that the list given in (Butler and Domeisen, 2021) contains only final warming events, rather than all warming events. Or consider omitting this part of the sentence

L131 Please consider mentioning that the dates of the SSW and SPV can be found later in Figures 7 and 8. I wanted to suggest adding a table with dates, but it seems excessive, since the information appears later in the text.

Fig 1c The model bias spans from -4.5 to 4.5% while the frequency itself changes from 0 to 45%, do you think that the bias is statistically significant in this case?

L154-155 Repetition of "in ERA-5"in the sentence, please remove one of them

L155 Did you look at the individual events before constructing the composite? It would be interesting to know which evens had stronger response. However, the washed-out signal in the reforecasts might show that the model underestimates the response, especially averaged over 28 days of forecast.

L196 "The statistical significance of *this* shift…" it is not clear whether you refer to the shift compared to all winter days, or the small shift of reforecasts compared to reanalysis

Fig. 4 As I understand, the figure shows cyclone frequency anomalies after the 14 events in each subplot, but in this case what does the height of the bars show? Counts on y-axis does not add up. If you used a somehow broader statistics, please clarify that in the caption.

Fig.5 and L212 Could you explain why there are more cyclone tracks (black lines) detected in reforecasts than in reanalysis? I suppose that you used each ensemble member separately rather than ensemble mean, which could be mentioned in text for easier understanding. Also, you mention in Data and Methods that in this part you use more reforecasts from three model versions, but could you explain more in detail why do you use other model versions. The temporal resolution increase to 6-hourly data is understandable here.

L248 Did you check this correspondence case-by-case here, rather than the overall ratio?

Fig.7b,d It would probably be better for understanding if you indicated in the figure that N=10 for enhanced cyclone frequency and N=4 for the reduced, etc.

L261 Did you have a look why the week-1 hit rate for 11 Feb 2005 was so low, especially considering that the skill is higher on the following weeks and the averaged skill is rather high (0.7 from Fig. 7c)?

L270 "…predicted a weakening of the cyclone frequency in the period that followed the SSW." As I understand the majority of ensemble members still predicted the increased cyclone frequency on week 1 and 2 in this case, so maybe you can specify that it is not about the period that directly follows the SSW.

L297 It could be worth specifying that in case of SSW it is about the reduced frequency

L308 Consider adding "after SSW events *in these cases*." as temperatures are not always predicted poorly after SSWs.

---

## Referee Comment (RC2)

**Manuscript Title:** Stratospheric influence on the winter North Atlantic storm track in subseasonal reforecasts

**Authors:** H. Afargan-Gerstman, D. Büeler, C. Ole Wulff, M. Sprenger, D. I. V. Domeisen

**Recommendation:** Reject but resubmit

**Summary**
The study examines changes in the North Atlantic storm track following either strong or weak stratospheric polar vortex conditions in both reanalysis and the ECMWF subseasonal model reforecasts. Using reanalysis, the authors illustrate that the storm track changes often follow the expected canonical response to each polar vortex state; i.e., an anomalously poleward-enhanced storm track following strong events and an anomalously equatorward-enhanced storm track following weak stratospheric polar vortex events. However, this is not always the case, and the operational models struggle to capture these contrary cases. Additionally, there is no significant change in the distribution of cyclone frequency after weak vortex conditions in the reforecasts or in reanalysis when compared to winter climatology. The only significant shift in cyclone frequency is detected following strong vortex states. In terms of predictability from the ECMWF, there is somewhat better predictability for getting the sign of the cyclone frequency anomaly correct following the strong vortex cases versus the weak ones, even in the non-canonical cases. Of course, the hit rate decreases with lead time (as might be expected), but it could still be useful for the strong vortex cases for a forecasting application.

**Overall Opinion**
The study presents some interesting results on how the state of the stratospheric polar vortex can offer some improved forecast skill for cyclone frequency across the North Atlantic (both intensity and spatial area). I recognize the work that the authors put into many of these analyses and appreciate some of the results, including the difference in forecast skill of storm tracks between strong and weak vortex states. Unfortunately, I do not find the results particularly novel; moreover, they ack the robustness and completeness that could make this work much more useful for the forecasting community. As such, I find the manuscript slightly incomplete in its presentation - there is more that would need to be completed to bolster the main messages of the work. Hence, I regretfully recommend that this paper not be published in its current form but instead sent back to the authors for further analysis and resubmitted at a later time. More detailed reasons for my decision are below.

**Reasons for Rejection**
First, the study examines only one subseasonal model (ECMWF) and therefore lacks a generalized view of how other leading subseasonal prediction systems reproduce the stratosphere-North Atlantic storm track relationship. The fields from the reforecasts of the other models are readily accessible and possible to be analyzed and compared/contrasted. I am not necessarily advocating using *every* model, but I think adding a few more will be very useful and strengthen the message.

Next, I found aspects of the methodology confusing. In the methods section, the authors mention that they use the ERA-Interim reanalysis product for determining the state of the

polar vortex (strong vs weak) but then use ERA5 for their analyses. Determination of events in the ERA5 dataset is very straightforward. So, to be consistent, the authors should use one reanalysis only throughout their work. Next, since the ERA5 is used to initialize the ECMWF reforecasts, and since the two share aspects of their modeling components, independence in the comparisons is hard to justify. Again, this aspect limits the applicability of the results of this work to other forecast systems and reanalyses. Next, the authors also comment frequently on the limited sample size from ERA5 for their results. This facet factors into their significance testing and other conclusions (e.g., Fig. 5). If sample size is too small, why should we trust the results? I am not saying that the limited sample size is a game-ender for the paper (trust me - this is a constant issue with my own work!). But, to use this concern over and over again in the manuscript as a caveat raises questions as to whether or not the findings are just an artifact of a short sample size.

Finally, I was disappointed that the paper did not investigate any physical reasoning for why the storm tracks change as they do in reanalysis vs the reforecasts. The authors mention a few times that their results are "consistent with" previous studies, which is good. But, the reforecasts and their multiple ensemble members offer a fantastic opportunity for the authors to address the "why." They could explore changes in wave fluxes, baroclinicity, jet stream dynamics, etc. and provide an idea of why the stratosphere is influencing the storm tracks the way it is. I think this is a missed opportunity with this paper, thus making it contribution less novel than it otherwise could be.

For these reasons, I must unfortunately recommend the paper not be published in its current form. Instead, I would suggest the authors consider my suggestions above, expand their study, and then resubmit the paper with these improvements. I think this avenue would be the most beneficial for the authors and for the subseasonal community.

**Other Comments**
1. **The acronym "SPV."** The use of this acronym is confusing - it is normally used to mean "stratospheric polar vortex" in many other papers. Furthermore, I don't find that the acronym is necessary in the work - "strong vortex events" is clear enough and not overly long. I recommend that the authors reconsider using this acronym.

2. **Lines 154-155.** I don't understand this sentence. How is the "response in ERA5"... "stronger in ERA5?"

3. **Line 221.** Either the results are statistically significant or they are not - they cannot be "partly significant."

4. **Lines 281-283.** Is this a "result" or "finding" that is unique to this work? I think that finding has already been shown in many past works and is also based on the fundamentals of what the jet stream is.

5. **Lines 291-293.** How would the authors propose to increase the sample size to meet their objective of determining the robustness of the results? (See my comments above as well.)

6. **Figures 2 and 3.** How is significance tested exactly for the reforecasts? What is the null hypothesis?

7. **Figure 7.** "Successfully" is spelled incorrectly in the y-axis labels of panels (b) and (d). Also, it is unclear what an "increase of cyclone frequency anomaly" means. Is it that it is a positive anomaly, or that the anomaly actually gets more positive over some time?

8. **Code and data availability.** The authors have not provided a public-accessible repository where their code is available. Please set up a Github and place your code on there for transparency and accessibility.

---

## Author Comment (AC1)

**Response to Reviewers - 1st revision**

Dear Editor,

We would like to thank all reviewers for their reviews of our manuscript and their insightful comments. Please find our detailed responses to the reviewers' comments and suggestions below.

The changes have been included into the manuscript (indicated in **bold**). All line numbers refer to the new (annotated) version of the manuscript.

Sincerely, The authors

**Reviewer 1:**

**Summary**

This analysis identified and examined the impact of stratospheric extremes on the extratropical storm track, cyclone frequency and cyclone life cycle characteristics at subseasonal time range. The results show that events with a canonical response after weak and strong stratospheric polar conditions are better predicted compared to the events without the canonical response. This possibly leads to over-prediction by the model as it tends to forecast the expected response. The predictability after SSW events is less uniform across events compared to the predictability of the surface response after SPV states. In my opinion, while the response analysis may be not very novel, it provides solid quantification. The most interesting part discusses the predictability of the response and provides new insights into the driving mechanisms. Therefore, I recommend this manuscript for publication, but I also suggest a few comments below.

**Major comments**

1. Had you considered to split the 28 days period into, for example, two parts (week 1-2 and week 3-4) in the first part of the manuscript? It would probably be interesting to look at least at U850 and cyclone frequency anomalies especially having in mind the results of the predictability part.

**We thank the reviewer for this comment. We have revised the manuscript and added a new figure (Figure 9) to demonstrate the time evolution of cyclone frequency response following extreme stratospheric events (SSWs and strong vortex events). In the new figure, we plot the time evolution of zonal mean cyclone frequency anomalies (ensemble mean) in the North Atlantic Basin (see box in Figure 3). To evaluate the uncertainty in the prediction of cyclone frequency after these events, we plot the average ensemble spread (superimposed in black contours). This additional analysis provides insight regarding the large latitudinal differences in the ensemble spread, suggesting a limited capability of the reforecasts in reproducing the response in mid-**

and high-latitudes. Particularly, the response in latitudes poleward of 55N are especially uncertain.
In addition, to address the dynamical aspect of extratropical cyclone prediction on subseasonal timescales, we consider both the time evolution of the jet stream and the storm track.

2. The box position choice does not seem well explained. You say that in this region the increase in cyclone frequency is biggest after SSW events (L185), but the anomalies are biggest only in reforecasts (Fig. 3a). Moreover, the anomalies are biggest and statistically significant over the Northern Europe in reanalysis (Fig. 3c), which can be also seen in reforecasts. Had you considered taking a box more to the north-east of its current position? Also, was your choice of the box position based only on the anomalies after the SSW events? I see that the biggest anomalies after the SPV cases are still concentrated inside the box (Fig. 3b), but maybe this can be pointed out in the text.

The boundaries of the box were determined according to the region of largest increase in cyclone frequency after SSW events (considering all 14 events in the dataset). We agree with the reviewer that a larger or shifted box can better capture the anomalies after both SSWs and SPV events. Therefore, we have shifted the southern boundary to 35°N and extended the northern boundary to 55°N. In the revised version, we focus our analysis on the mid-latitude region (35°-55°N) of the North Atlantic (60°W-0°E). This region, located on the southern flank of the North Atlantic storm track, is where the change in cyclone frequency after SSW and strong polar vortex events is the largest. We have updated all the plots in the manuscript according to the new box definition.

**Minor comments**

L32 "predication" -¿ prediction
Corrected.

L34 Add brackets to citation
Corrected.

L48-51 This sentence seems to repeat the information given above, please consider removing it or rephrasing the repetition
We have rephrased that paragraph as suggested by the reviewer to avoid repetition.

| Dataset/Forecast | Operation period |
|:---:|:---:|
| Cycle 46R1 | from 11/06/2019 |
| Cycle 47R1 | from 30/06/2020 |
| Cycle 47R2 | from 11/05/2021 |
| Cycle 47R3 | from 13/10/2021 |

Table 1: Implementation dates of each ECMWF model version. Source: `https://confluence.ecmwf.int/display/S2S/ECMWF+Model`.

L88 As reforecasts are initialized in conjunction with real-time forecasts, could you provide here the dates/years of the real-time forecasts? You indicate below the model versions used, but this is potentially confusing, as, for example, the 46R1 version does not have reforecasts for December 2019 2.

**The ensemble re-forecasts consist of a 11-member ensemble starting the same day and month as a real-time forecast (Monday and Thursday), covering the past 20 years. The implementation dates of each ECMWF model version is summarized in the table below (Table 1). For example, the reforecasts of December 2, 2019 belongs to the model cycle CY46R1 since it has been computed between 11/06/2019 and 30/06/2020. The reforecast for this date has been initialized on same date as the real-time forecast of 02/01/2020. This reforecast consisted of a 11-member ensemble starting on 2nd January 2000, 2nd January 2001,... to 2nd January 2019 (20 years).**

**To clarify this point, we have added this information to the Methods section (lines 80-85).**

L97 Please consider adding "... in the ECMWF model and in reanalysis..." if you used the same algorithm

**Corrected.**

L100 Consider adding here a remark that the number of the cyclone tracks can be found in Fig.6

**Corrected.**

L104 DJF -¿ DJFM for consistency throughout the text

**Corrected.**

L108 Did you use cross validation when computing the anomalies for each ensemble member?

**Cyclone frequency anomaly for each ensemble member is computed as the difference in the number of cyclones detected in the 28 days after the SSW and the climatological cyclone frequency for this period. As the computation of these anomalies is**

mathematically straightforward (i.e., an anomaly is defined as a deviation from the climatological mean), we have not performed cross validation when computing the anomalies.

L108 While I understand the choice of 28 days, it could be better clarified here for better understanding
**We have added a clarification in the text regrading the choice of the 28 days period, as follows (lines 107-111):**
**In the NH, anomalies in the tropospheric circulation after extreme stratospheric events (such as SSWs, weak vortex events and strong vortex events) can persist for up to 60 days after their onset, and thus may prove to be useful for tropospheric weather prediction. A period of 28 days after the onset of SSWs and strong vortex events is chosen in order to understand the initial tropospheric response and its potential for subseasonal predictions of the surface response.**

L117 I wonder of you checked if there is no difference indeed when using ERA-Interim or ERA-5?
**We have repeated the analysis (detection of SSW and SPV events) using ERA-5 data instead of ERA-Interim, and the results remain the same. We have updated the manuscript such that the entire analysis is performed using ERA-5 reanalysis.**

L120 Please specify that the list given in (Butler and Domeisen, 2021) contains only final warming events, rather than all warming events. Or consider omitting this part of the sentence
**We rephrased this part of the sentence to clarify this point.**

L131 Please consider mentioning that the dates of the SSW and SPV can be found later in Figures 7 and 8. I wanted to suggest adding a table with dates, but it seems excessive, since the information appears later in the text.
**A full list of SSW and SPV dates that are used in this study can be found in Figure 7. We added this information to the text (lines 130-132).**

Fig 1c The model bias spans from -4.5 to 4.5% while the frequency itself changes from 0 to 45%, do you think that the bias is statistically significant in this case?
**Following the reviewer's comment, we have added an analysis of the statistical significance of the cyclone frequency bias compared to reanalysis in Fig. 1.**

L154-155 Repetition of "in ERA-5" in the sentence, please remove one of them
**Corrected.**

L155 Did you look at the individual events before constructing the composite? It would be interesting to know which events had stronger response. However, the washed-out signal in the reforecasts might show that the model underestimates the response, especially averaged over 28 days of forecast.

**The variability of extratropical cyclone response to SSW and strong vortex events is examined in the revised manuscript (section 3.5) through an analysis of individual events. For this purpose, we take the following approach: first, we analyze the reforecast performance in predicting the regional cyclone frequency after stratospheric events.**

**Second, we added an analysis of model spread as a function of latitude and time. This allows us to examine how changes in cyclone frequency evolve with time, during the 28-day period of the SSW and strong vortex events. We find that the model spread (represented as the standard deviation of an ensemble) is relatively small close to the event onset, and becomes larger in weeks 3-4 and in high-latitudes (55°N to 65°N).**

L196 "The statistical significance of this shift..." it is not clear whether you refer to the shift compared to all winter days, or the small shift of reforecasts compared to reanalysis

**We refer to statistical significance of the shift in the distribution of cyclone frequency anomalies in the case of strong polar vortex events relative to all winter days. We rephrased this paragraph (lines 195-200) to clarify this point.**

Fig. 4 As I understand, the figure shows cyclone frequency anomalies after the 14 events in each subplot, but in this case what does the height of the bars show? Counts on y-axis does not add up. If you used a somehow broader statistics, please clarify that in the caption.

**For each stratospheric events (SSW or strong polar vortex event), we analyze the distribution of the cyclone frequency anomaly in the selected region following the event. Anomalies are averaged over a period of 28 days (days 1-28 with respect to the central date of the stratospheric event). Since we have 14 events for each event type, the total number of counts is [N=14] for ERA5 reanalysis and [N=140] for the reforecasts (14 events x 10 ensemble members]).**

**The left y-axis in Figure 4 shows the probability density, hence each bin displays the bin's count, divided by the total number of counts and the bin width, so that the area under the histogram integrates to 1. Displaying the probability density allows a direct comparison between the distribution of anomalies in the reanalysis and in the reforecasts. We have clarified this information in Figure 4's caption, and changed the y-axis label from "counts" to "probability density".**

**For comparison, a histogram of cyclone frequency anomalies following SSW and strong**

[Figure]

Figure 1: Histogram of cyclone frequency anomalies (in counts) following (a,c) SSW and (b,d) strong polar vortex events in ERA5 reanalysis (grey) and in the ECMWF reforecasts (purple). Anomalies are averaged over the mid-latitude North Atlantic. The grey curve in each panel indicates the climatological probability density for all days in DJFM in the reforecasts.

**polar vortex events in terms of raw counts for both reanalysis (panels a,b) and refore-casts (c,d) is shown below (Figure 1).**

Fig.5 and L212 Could you explain why there are more cyclone tracks (black lines) detected in reforecasts than in reanalysis? I suppose that you used each ensemble member separately rather than ensemble mean, which could be mentioned in text for easier understanding. Also, you mention in Data and Methods that in this part you use more reforecasts from three model versions, but could you explain more in detail why do you use other model versions. The temporal resolution increase to 6-hourly data is understandable here.

**We have added this information to the text to address these important points (lines 212-215). There are more tracks in reforecasts than in reanalysis due to the use of all available ensemble members (11 members) rather than the ensemble mean.**

Regarding the use of three different model version, this is a result of implementation dates for each new prediction system version (cycle) by the ECMWF. We have included the table with the specific dates (Table 1 in the response). Depending on the time in which data has been downloaded, the model version of the reforecast will be different.

L248 Did you check this correspondence case-by-case here, rather than the overall ratio?
**Yes, the ratio of SSW events with and without a 'canonical' downward response is based on case-by-case examination of the response, as shown in Figure 7. In the revised manuscript, this ratio has slightly changed from the previous manuscript as a result of the change in the region of averaging (see new box position in Figure 3).**

Fig.7b,d It would probably be better for understanding if you indicated in the figure that N=10 for enhanced cyclone frequency and N=4 for the reduced, etc.
**We have now added this information to the plot (see x-axis labels).**

L261 Did you have a look why the week-1 hit rate for 11 Feb 2005 was so low, especially considering that the skill is higher on the following weeks and the averaged skill is rather high (0.7 from Fig. 7c)?
**The reasons for the change in skill for the 11 Feb 2005 SSW events are yet unresolved. Analyzing the potential causes requires a more focused study on the dynamics of this specific event, as was done, for instance, for the 2018 SSW event (e.g., Karpechko et al., 2018, Kautz et al., 2020). While this is an interesting question, we believe this analysis is out of the scope of this paper.**

L270 "...predicted a weakening of the cyclone frequency in the period that followed the SSW." As I understand the majority of ensemble members still predicted the increased cyclone frequency on week 1 and 2 in this case, so maybe you can specify that it is not about the period that directly follows the SSW.
**Thanks for this comment, we rephrased this sentence to specify that we refer to week 1 and 2 in this case, and not for the entire period.**

L297 It could be worth specifying that in case of SSW it is about the reduced frequency
**We have rephrased that sentence to clarify that.**

L308 Consider adding "after SSW events in these cases." as temperatures are not always predicted poorly after SSWs.
**We corrected that.**

**1   Reviewer 2**

**Summary**

The study examines changes in the North Atlantic storm track following either strong or weak stratospheric polar vortex conditions in both reanalysis and the ECMWF subseasonal model reforecasts. Using reanalysis, the authors illustrate that the storm track changes often follow the expected canonical response to each polar vortex state; i.e., an anomalously poleward- enhanced storm track following strong events and an anomalously equatorward-enhanced storm track following weak stratospheric polar vortex events. However, this is not always the case, and the operational models struggle to capture these contrary cases. Additionally, there is no significant change in the distribution of cyclone frequency after weak vortex conditions in the reforecasts or in reanalysis when compared to winter climatology. The only significant shift in cyclone frequency is detected following strong vortex states. In terms of predictability from the ECMWF, there is somewhat better predictability for getting the sign of the cyclone frequency anomaly correct following the strong vortex cases versus the weak ones, even in the non-canonical cases. Of course, the hit rate decreases with lead time (as might be expected), but it could still be useful for the strong vortex cases for a forecasting application.

**Overall Opinion**

The study presents some interesting results on how the state of the stratospheric polar vortex can offer some improved forecast skill for cyclone frequency across the North Atlantic (both intensity and spatial area). I recognize the work that the authors put into many of these analyses and appreciate some of the results, including the difference in forecast skill of storm tracks between strong and weak vortex states. Unfortunately, I do not find the results particularly novel; moreover, they lack the robustness and completeness that could make this work much more useful for the forecasting community. As such, I find the manuscript slightly incomplete in its presentation - there is more that would need to be completed to bolster the main messages of the work. Hence, I regretfully recommend that this paper not be published in its current form but instead sent back to the authors for further analysis and resubmitted at a later time. More detailed reasons for my decision are below.

**1.1   Reasons for Rejection**

First, the study examines only one subseasonal model (ECMWF) and therefore lacks a generalized view of how other leading subseasonal prediction systems reproduce the stratosphere-North Atlantic storm track relationship. The fields from the reforecasts of the other models are readily accessible and possible to be analyzed and compared/contrasted. I am not necessarily advocating using every model, but I think adding a few more will be very useful and strengthen the message.

We thank the reviewer for this comment. Indeed, we agree that a systematic analysis of model biases in the downward impact of extreme stratospheric events across a wide range of subseasonal forecast systems would be an important step towards a better understanding of the role of the stratosphere for prediction of surface climate on subseasonal to seasonal timescales.

However, inter-comparison studies of S2S prediction are way more complex, and require more effort (computationally, as well as time-wise), more data, and usually are done as large community/collaborative studies, as was recently done, for instance, in the Lawrence et al., 2022 for a systematic analysis of model biases in the stratosphere. In the context of extratropical cyclone analysis, this would require implementation of the cyclone detection algorithm for identification and tracking of extratropical cyclones across all prediction systems available in the S2S project.

One of the reasons for using the ECMWF prediction system in this study is due to its more highly resolved stratosphere (relative to other models in the S2S project, e.g., Domeisen et al., 2020). Overall, the ECMWF model has been shown to have a good representation of the variability in the stratospheric polar vortex, in terms of extreme event magnitude and the associated dynamical drivers (Wu et al., 2022). Furthermore, the model represents mid-latitude storm track well, as shown in Fig. 1 in the revised manuscript.

We would be happy to perform a more complex, extended study on the S2S biases in the prediction of the storm track in the future, as a follow up on the current manuscript and its main findings using the ECMWF extended-range prediction system.

References:

- Domeisen, D. I., Butler, A. H., Charlton-Perez, A. J., Ayarzagüena, B., Baldwin, M. P., Dunn-Sigouin, E., ... and Taguchi, M. (2020). The role of the stratosphere in subseasonal to seasonal prediction: 2. Predictability arising from stratosphere-troposphere coupling. Journal of Geophysical Research: Atmospheres, 125(2), e2019JD030923. `https://doi.org/10.1029/2019JD030923`

- Lawrence, Z. D., Abalos, M., Ayarzagüena, B., Barriopedro, D., Butler, A. H., Calvo, N., de la Cámara, A., Charlton-Perez, A., Domeisen, D. I. V., Dunn-Sigouin, E., García-Serrano, J., Garfinkel, C. I., Hindley, N. P., Jia, L., Jucker, M., Karpechko, A. Y., Kim, H., Lang, A. L., Lee, S. H., Lin, P., Osman, M., Palmeiro, F. M., Perlwitz, J., Polichtchouk, I., Richter, J. H., Schwartz, C., Son, S.-W., Statnaia, I., Taguchi, M., Tyrrell, N. L., Wright, C. J., and Wu, R. W.-Y.: Quantifying stratospheric biases and identifying their potential sources in subseasonal forecast systems, Weather Clim. Dynam., 3, 977–1001, 2022. `https://doi.org/10.5194/wcd-3-977-2022`

- Wu, R. W. Y., Wu, Z., and Domeisen, D. I. (2022). Differences in the sub-seasonal predictability of extreme stratospheric events. Weather and Climate Dynamics, 3(3), 755-776. `https://doi.org/10.5194/wcd-3-755-2022`

Next, I found aspects of the methodology confusing. In the methods section, the authors mention that they use the ERA-Interim reanalysis product for determining the state of the polar vortex (strong vs weak) but then use ERA5 for their analyses. Determination of events in the ERA5 dataset is very straightforward. So, to be consistent, the authors should use one reanalysis only throughout their work.

**We agree with the reviewer's comments regrading a consistent use of ERA-5 throughout the paper, and therefore we have adjusted the methods section accordingly. In the revised version, the stratospheric extreme events have been detected using the ERA-5 data (which allows a direct comparison with the dates detected by ERA-Interim). Using ERA-5 for events detection does not change the results of this paper.**

Next, since the ERA5 is used to initialize the ECMWF reforecasts, and since the two share aspects of their modeling components, independence in the comparisons is hard to justify. Again, this aspect limits the applicability of the results of this work to other forecast systems and reanalyses.

**Although the ECMWF reforecasts are initialized from ERA5 data, they evolve from the reanalysis with time. Therefore, independence in the comparison is not needed for the verification of the forecasts. This study focuses on the ECMWF model from the reasons described in the previous answers, which allows an analysis of the model bias in the 4-weeks following the forecast initialization.**

Next, the authors also comment frequently on the limited sample size from ERA5 for their results. This facet factors into their significance testing and other conclusions (e.g., Fig. 5). If sample size is too small, why should we trust the results? I am not saying that the limited sample size is a game-ender for the paper (trust me - this is a constant issue with my own work!). But, to use this concern over and over again in the manuscript as a caveat raises questions as to whether or not the findings are just an artifact of a short sample size.

**We thank the reviewer for pointing out this important topic. Indeed, studies of atmospheric variability, and stratospheric variability in particular, are limited by the small sample size of the observational record. One possible way to increase the sample size is using atmospheric models, that are run for longer time periods compared to the historical record, and thus able to produce a larger number of events (e.g., Afargan et al., 2022).**

**The extent to which we can assess biases is limited by the sample size (e.g., Lawrence et al., 2022; Domeisen et al., 2020). Despite a relatively small sample size (as in this study), these studies are able to assess model biases in the stratosphere across a wide**

range of subseasonal forecast systems when the results do show evidence of a systematic bias.

**In the revised manuscript, we emphasize the variability among the events and its implications on event predictability. Focusing on the inter-variability of these events, given the small sample size limitation, complements the results shown by the composites of zonal wind and cyclone frequency (e.g., Figures 2 and 3) and provides a more detailed perspective on the impact of these events - without making any preliminary assumptions on their systematic bias. In this way, we are able to take a more careful path and overcome the limitation of the historical record.**

References:

- Afargan-Gerstman, H., Jiménez-Esteve, B., and Domeisen, D. I. (2022). On the Relative Importance of Stratospheric and Tropospheric Drivers for the North Atlantic Jet Response to Sudden Stratospheric Warming Events. Journal of Climate, 35(19), 2851-2865.

- Domeisen, D. I., Butler, A. H., Charlton-Perez, A. J., Ayarzagüena, B., Baldwin, M. P., Dunn-Sigouin, E., ... and Taguchi, M. (2020). The role of the stratosphere in subseasonal to seasonal prediction: 2. Predictability arising from stratosphere-troposphere coupling. Journal of Geophysical Research: Atmospheres, 125(2), e2019JD030923. `https://doi.org/10.1029/2019JD030923`

- Lawrence, Z. D., Abalos, M., Ayarzagüena, B., Barriopedro, D., Butler, A. H., Calvo, N., de la Cámara, A., Charlton-Perez, A., Domeisen, D. I. V., Dunn-Sigouin, E., García-Serrano, J., Garfinkel, C. I., Hindley, N. P., Jia, L., Jucker, M., Karpechko, A. Y., Kim, H., Lang, A. L., Lee, S. H., Lin, P., Osman, M., Palmeiro, F. M., Perlwitz, J., Polichtchouk, I., Richter, J. H., Schwartz, C., Son, S.-W., Statnaia, I., Taguchi, M., Tyrrell, N. L., Wright, C. J., and Wu, R. W.-Y.: Quantifying stratospheric biases and identifying their potential sources in subseasonal forecast systems, Weather Clim. Dynam., 3, 977–1001, 2022. `https://doi.org/10.5194/wcd-3-977-2022`

Finally, I was disappointed that the paper did not investigate any physical reasoning for why the storm tracks change as they do in reanalysis vs the reforecasts. The authors mention a few times that their results are "consistent with" previous studies, which is good. But, the reforecasts and their multiple ensemble members offer a fantastic opportunity for the authors to address the "why." They could explore changes in wave fluxes, baroclinicity, jet stream dynamics, etc. and provide an idea of why the stratosphere is influencing the storm tracks the way it is. I think this is a missed opportunity with this paper, thus making it contribution less novel than it otherwise could be. **We thank the reviewer for these suggestions. Predictability of the downward impact after extreme stratospheric events strongly differs among events, even of the**

[Figure]

Figure 2: (a-d) Zonal mean zonal wind and (e-h) cyclone frequency anomalies following (left) SSW and (right) strong vortex events in the ECMWF reforecasts (purple). Anomalies are averaged over the mid-latitude North Atlantic (black box in Figure 3). The grey curve in each panel indicates the ensemble spread.

same type (e.g., Domeisen et al., 2020; Wu et al., 2022). The reasons for the observed differences in the predictability are not yet resolved, and often require an analysis from a case-by-case perspective, as was done for example for the 2018 SSW events (e.g., Karpechko et al., 2018, Kautz et al., 2020).

Below, we analyse the difference in predictability between SSW and strong polar vortex events for events with the expected (i.e., canonical) downward response. An "expected" response is defined here as positive (negative) cyclone frequency anomalies in midlatitudes following SSW events (strong vortex events). We find that the "expected" cyclone frequency response after SSW events is reproduced well by ensemble members that also capture the positive anomalies in zonal wind at 805 hPa, whereas the "unexpected" cyclone frequency response is consistent with negative zonal wind anomalies. While the "unexpected" response is wrong in the first days after the event onset, and becomes "expected", i.e., positive from day 15, this still affects the entire 28-day period. This analysis sheds light on our conclusions presented in the manuscript, regarding the importance of inter-event variability in the predictability after stratospheric events.

References:

- Domeisen, D. I., Butler, A. H., Charlton-Perez, A. J., Ayarzagüena, B., Baldwin, M. P., Dunn-Sigouin, E., ... and Taguchi, M. (2020). The role of the stratosphere in subseasonal to seasonal

prediction: 2. Predictability arising from stratosphere-troposphere coupling. Journal of Geophysical Research: Atmospheres, 125(2), e2019JD030923. `https://doi.org/10.1029/2019JD030923`

- Karpechko, A. Y., Charlton-Perez, A., Balmaseda, M., Tyrrell, N., and Vitart, F. (2018). Predicting sudden stratospheric warming 2018 and its climate impacts with a multimodel ensemble. Geophysical Research Letters, 45(24), 13-538. `https://doi.org/10.1029/2018GL081091`.

- Kautz, L. A., Polichtchouk, I., Birner, T., Garny, H., and Pinto, J. G. (2020). Enhanced extended-range predictability of the 2018 late-winter Eurasian cold spell due to the stratosphere. Quarterly Journal of the Royal Meteorological Society, 146(727), 1040-1055. Wu, R. W. Y., Wu, Z., and Domeisen, D. I. (2022). Differences in the sub-seasonal predictability of extreme stratospheric events. Weather and Climate Dynamics, 3(3), 755-776. `https://doi.org/10.5194/wcd-3-755-2022`

**Other Comments**

1. The acronym "SPV." The use of this acronym is confusing - it is normally used to mean "stratospheric polar vortex" in many other papers. Furthermore, I don't find that the acronym is necessary in the work - "strong vortex events" is clear enough and not overly long. I recommend that the authors reconsider using this acronym.

**The use of the acronym "SPV" for "strong polar vortex" can be found in the litrature (e.g., Oehrlein et al., 2020, Díaz-Durán et al., 2017). However, to avoid confusion due to the different uses of this acronym, we corrected "SPV" to "a strong vortex event" throughout the manuscript.**

2. Lines 154-155. I don't understand this sentence. How is the "response in ERA5"... "stronger in ERA5?"

**We rephrased this paragraph and removed this sentence.**

3. Line 221. Either the results are statistically significant or they are not - they cannot be "partly significant."

**Thank you for pointing this out. We have rephrased that sentence to clarify that the results in that case (difference in cyclone intensity) are not significant in ERA5 (lines 222-223).**

4. Lines 281-283. Is this a "result" or "finding" that is unique to this work? I think that finding has already been shown in many past works and is also based on the fundamentals of what the jet stream is.

**We rephrased this paragraph to emphasize that our analysis is consistent with previous studies, in context of the expected stratospheric impact. However, we emphasize a**

possible overconfidence of the model with respect to reanalysis to predict the canonical response after SSW events - a topic which has received less attention in the literature.

5. Lines 291-293. How would the authors propose to increase the sample size to meet their objective of determining the robustness of the results? (See my comments above as well.)
**One option for increasing the sample size is to use a modeling study, hence to run a model for a longer time and generate more SSW events. However, such procedure is less relevant when analyzing reforecasts, as done in the current study. An additional approach is to perturb the stratosphere**

6. Figures 2 and 3. How is significance tested exactly for the reforecasts? What is the null hypothesis? **In Figures 2 and 3 significance is tested based on a Student's t-test. The null hypothesis (H0) is there is no difference between the means of these two variables (i.e., zonal wind anomalies and 0). Significance is tested for each grid point. An additional and more detailed significance testing is performed in section 3.4, where we investigate how the average cyclone life cycle characteristics depend on the extreme states of the stratospheric polar vortex. In Figure 5 and Figure 6, the confidence interval is obtained from a bootstrapped distribution of median latitudes (based on 1000 random resamples of the tracks with replacement).**

7. Figure 7. "Successfully" is spelled incorrectly in the y-axis labels of panels (b) and (d). Also, it is unclear what an "increase of cyclone frequency anomaly" means. Is it that it is a positive anomaly, or that the anomaly actually gets more positive over some time?
**We have corrected the typo. As for the meaning of "increase of cyclone frequency anomaly", this term refers to a positive anomaly in cyclone frequency. Red bars in Fig. 7a indicate the proportion of ensemble members that show an average increase in cyclone frequency over the selected region, whereas blue bars indicate a decrease.**

8. Code and data availability. The authors have not provided a public-accessible repository where their code is available. Please set up a Github and place your code on there for transparency and accessibility.
**We thank the reviewer for this comment. We have created a public-accessible Github repository for the code and datasets (`https://github.com/hillaag/downward_impact_analysis_tools_for_S2S.git`). We now specify this information under the Code Availability section.**

---

## Referee Report (RR1)

**Review: Stratospheric influence on the winter North Atlantic storm track in subseasonal reforecasts**

Hilla Afargan-Gerstman, Dominik Büeler, C. Ole Wulff, Michael Sprenger, and Daniela I.V. Domeisen

**General comment**

I thank the authors for thoroughly replying to all the reviewers' comments and including some further analyses / figures, which in my opinion helped to improve the manuscript. I agree with the authors on the use of the ECMWF model only as a similar systematic analysis using more/all of the S2S models would require much more effort and it would result in a much longer manuscript. Moreover, such inter-comparison study can be done as a continuation of the present work. I therefore only have a few minor comments left, after which I see the manuscript ready to be published.

**Minor comments**

Figure 3 There are currently two black boxes which is confusing throughout the text. I suggest distinguishing between them, for example, using another color or a dashed line for one of them.

L198 As you now have boxes plotted in all of the subplots it's enough to say 'Figure 3'. However, it is good to specify here which box you're referring to (see my comment above).

Figure 8 caption: although you mention the coordinates, it's better to specify here which box you're referring to once you've distinguished the boxes (see my comment above).

Figure 10 Please mention in the caption that this is a 28d average

L342 Although you mention using Z100 in the Data and Methods, I believe that this is the first time you mention Z'100 (Z100 *anomalies*) so it makes sense to specify it here.

L358 please fill in the Figure number instead of '??'

L370 typo 'to **be** associated'

L373 probably 'to SSW event**s**'

L452 I think that you forgot to add the link to Github in the Code availability section.

---

## Referee Report (RR2)

**Manuscript Title:** Stratospheric influence on the winter North Atlantic storm track in subseasonal reforecasts

**Authors:** H. Afargan-Gerstman, D. Büeler, C. Ole Wulff, M. Sprenger, D. I. V. Domeisen

**Recommendation:** Major revisions

Overall Opinion
This is my second review of the manuscript. The prior version of the manuscript lacked two key elements that I thought made it unsuitable for publication: (1) The study focused on only one model, whereas the database has several other models with hindcasts that could be explored for the same phenomenon; and (2) the paper lacked any dynamical insight into *why* the results were the way they were. The authors responded that doing the analysis for more than one model would be too laborious and that the ECMWF model was a well-trusted subseasonal model. I *kind of* agree with this thought, though I think the paper would be strengthened with more than one model analyzed for the study. But, I won't hold up publication of the study based just on that issue. For the second point, the authors expanded their analysis to look at the nature of downward propagation and the tropospheric circulation after a strong or weak vortex event. Specifically, the authors contrast between ensemble members which correctly predict the anomaly of the North Atlantic cyclone frequency following a strong/weak polar vortex event and those that do not. However, I don't think the methodology used actually addresses the dynamical interpretation of the results that the authors intend.

So, taken together, I would consider the paper ready for publication but only after **major revisions,** particularly to the dynamical interpretation portion of the paper.

Major Revisions
1. **Sampling Issues and Figure 4.** The authors continue to note that sample size is a concern for many of their results, and I agree with this point. However, one area this is not addressed enough is in **Figure 4.** In particular, the sample size between reanalysis and reforests for the SSW and strong vortex events is about a factor of 10 different, which makes comparison of the probability distribution functions (PDFs) very difficult (in fact, I question the reproducibility and representation of a distribution of a variable with only 14 samples). For example, the authors could repeatedly sample 14 random cases (with replacement) from the reforecasts to make a comparative PDF with reanalysis. I think in its current form, it is hard to argue about statistically significant differences in these distributions, whether comparing the cases or comparing reanalysis to reforecasts. The authors may want to pursue alternate strategies to strengthen this argument in the paper.

2. **Dynamical Interpretation.** I like the efforts that the authors made in trying to bring some dynamical insight into their model evaluation study of Atlantic storm track changes due to stratospheric polar vortex variability. However, I am not convinced that the analyses shown actually accomplish this effort. In particular, the authors use mean sea level pressure (MSLP) to represent the tropospheric circulation changes after "successful" and "unsuccessful" forecasts. However, storm tracks are defined in this study using MSLP. So, it is a bit circular to argue that differences in MSLP ("the tropospheric circulation") are the

leading reason why there are changes in storm tracks (which are determined by MSLP). I like the use of lower tropospheric winds and even looking at the lower stratosphere (Z100). But, the authors should reconsider how they measure the tropospheric circulation and consider other variables for that other than MSLP.

**Minor Revisions**

1. **Lines 14-18.** This sentence is long and confusing to understand. Please revise.

2. **Lines 74-75.** The acronym "ERA5" already contains the word "reanalysis" in it. So, it is redundant to say "ERA5 reanalysis."

3. **Line 228.** Please move the comma from after "ERA5" to after "reforecasts."

4. **Lines 313.** There is no need to define "MSLP" again here. Also, I think it is unnecessary to introduce another acronym into the paper for the 100 hPa geopotential height anomalies. Instead, the authors can just use the already-defined acronym for 100 hPa geopotential height anomalies (Z100). Can you just write "Z100 anomalies?"

5. **Lines 338-340.** This sentence structure (with the parentheses) is no longer favored in journal articles for readability and understanding. Please rephrase as two sentences or in another way. Same comment for **Lines 411-412.**

6. **Line 353.** It looks like there is a missing figure reference here ("??").

7. **Lines 362-363.** I am unclear what "larger natural variability" means, particularly in reference to model reforecasts. What does "natural variability" in a simulated atmosphere mean?

8. **Line 364.** "...found to b associated with a..." —> "...found to be associated with a..."

9. **Lines 445-448.** The authors previously mentioned that they were going to have a Github site to make their data publicly accessible. This site is not listed here - please add the information for completion.

---

## Author Response (AR2)

**Response to Reviewers - 1st revision**

Dear Editor,

We would like to thank all reviewers and the editor for their reviews of our manuscript and their insightful comments. Please find our detailed responses to the reviewers' comments and suggestions below. The changes have been included into the manuscript (indicated in **bold**). All line numbers refer to the new (annotated) version of the manuscript.

In the revised version we have made several improvements, as follows:

- We have included a new analysis on the dynamical aspects of the downward impact following extreme stratospheric events (Section 3.7; Figures 10, 11 and 12), focusing on the difference between successful and unsuccessful forecasts.

- We have added a new analysis of the temporal changes in the MSLP and 850-hPa zonal wind anomalies after successful and unsuccessful predictions of the downward response.

- We have included an analysis of cyclone frequency anomalies over Europe after SSW and strong vortex events, as well as over the North Atlantic.

- We have corrected Figure 2 to show the 850-hPa zonal wind anomalies in all panels.

- We added a new analysis of temporal evolution of the ensemble spread (Figure 13).

Sincerely,

The authors

**1 Editor's review:**

**Major comments:**

1. I appreciate the use of ERA5 and additional discussion on differences between the successful and unsuccessful re-forecasts. Nevertheless, while I understand your limitations in using more datasets, I reckon adding at least one other model could have been valuable following a suggestion by Reviewer#2.

**Thank you for this suggestion. We agree that performing a multi-model study on the biases and predictability of extratropical storms in the North Atlantic and Europe following extreme stratospheric conditions is indeed an interesting and appealing idea relevant for a wide audience. However, the ECMWF model has been used as a single forecasting system in a wide number of studies on sub-seasonal and seasonal predictability (we have listed a few examples below: Lee et al., 2022, Winters 2021, Pyrina and Domeisen, 2023). Particularly, the ECMWF model has been used in**

various studies on stratospheric predictability and stratosphere-troposphere coupling. For SSWs, for instance, the ECMWF system is among the two models that have the smallest errors within a lead time of 2 weeks (Lawrence et al., 2022). Hence, for a process study and for this initial evaluation of a new aspect of stratosphere - troposphere coupling, using a single model, as it is done here, is the common pathway for scientific studies on this topic. Adding additional models would require us to approximately double the number of figures in the manuscript, and the choice of the model would be rather arbitrary given the wide range of possible choices, as there is no obvious second model to add. This addition would also require us to add a significant amount of text documenting the model biases and comparing the models with each other.

We agree, however, that model comparison studies are very worthwhile, with a focus on comparing model biases. In fact, a study on quantifying stratospheric biases and identifying their potential sources in subseasonal forecast systems, including a systematic analysis of model biases in the stratosphere across a wide range of subseasonal forecast systems, has recently been published in the Weather and Climate Dynamics journal (Lawrence et al., 2022). A follow-up study by the same authors is currently in preparation that provides a systematic review of the model biases in the same models with respect to the downward coupling from the stratosphere that can subsequently impact the tropospheric circulation across the globe. This study will however not cover the storm tracks in detail, and hence, a follow-up study is planned by the leading co-authors of this paper that will focus on a multi-model comparison of the stratospheric downward impact on the storm tracks using the ensemble hindcast data from the S2S Prediction Project Database (Vitart et al., 2017). We believe that such a study, that will focus mainly in the jet stream and storm track response, can be an important step towards model improvement on subseasonal-to-seasonal timescales and a better understanding the role of the stratosphere for weather and climate prediction.

**References:**

- Kolstad, E. W., Wulff, C. O., Domeisen, D. I., & Woollings, T. (2020). Tracing North Atlantic Oscillation forecast errors to stratospheric origins. Journal of Climate, 33(21), 9145-9157. `https://doi.org/10.1175/JCLI-D-20-0270.1`.

- Lawrence, Z. D., Abalos, M., Ayarzagüena, B., Barriopedro, D., Butler, A. H., Calvo, N., ... & Wu, R. W. Y. (2022). Quantifying stratospheric biases and identifying their potential sources in subseasonal forecast systems. Weather and Climate Dynamics, 3(3), 977-1001. `https://doi.org/10.5194/wcd-3-977-2022`.

- Lee, S. H., Charlton-Perez, A. J., Woolnough, S. J., & Furtado, J. C. (2022). How do

stratospheric perturbations influence North American weather regime predictions?. Journal of Climate, 35(18), 5915-5932. `https://doi.org/10.1175/JCLI-D-21-0413.1`.

- Pyrina, M., & Domeisen, D. I. (2023). Subseasonal predictability of onset, duration, and intensity of European heat extremes. Quarterly Journal of the Royal Meteorological Society, 149(750), 84-101. `https://doi.org/10.1002/qj.4394`.

- Vitart, F., Ardilouze, C., Bonet, A., Brookshaw, A., Chen, M., Codorean, C., ... & Zhang, L. (2017). The subseasonal to seasonal (S2S) prediction project database. Bulletin of the American Meteorological Society, 98(1), 163-173. `https://doi.org/10.1175/BAMS-D-16-0017.1`.

- Winters, A. C. (2021). Subseasonal prediction of the state and evolution of the North Pacific jet stream. Journal of Geophysical Research: Atmospheres, 126(17), e2021JD035094. `https://doi.org/10.1029/2021JD035094`.

2. One additional comment that I have on this analysis is that not all observed anomalous polar vortex events are followed by a 'canonical' response in cyclone activity. For example, if only 8(10) out of 14 strong polar vortex events are followed by weakening of cyclone activity over Europe (North Atlantic), is it reasonable to expect that if a 'canonical' response is found in ERA5 then all reforecasts should also demonstrate the canonical response? Following on from that, I think it might be valuable if Fig. 9 indicated ERA5 cyclone anomalies by week unless 'canonical' response held for all 4 weeks (please change 'SVs' to strong polar vortex events' for consistency).

**We thank the reviewer for this comment. Stratospheric variability is a potential source of atmospheric predictability on subseasonal and seasonal timescales (e.g., Baldwin et al., 2003; Maycock et al., 2011; Sigmond et al., 2013; Butler et al., 2018; Gerber et al., 2012; Scaife et al., 2016; Domeisen et al., 2019). Extreme events in the stratosphere, such as SSWs and strong vortex events, can have a persistent lower-stratospheric impact that often reaches the surface, where it can lead to changes in tropospheric variability on subseasonal timescales.**

**However, as only two-thirds of SSW events are followed by a 'canonical' downward impact (e.g., Afargan-Gerstman and Domeisen, 2020), predicting a 'canonical' downward impact after extreme stratospheric events depends on their event-to-event variability. The ECMWF forecast system have been found to overestimate the persistence of the negative NAO response following a weak polar vortex and the positive NAO response following a strong polar vortex (Kolstad et al., 2020). Consistent with that, we show that the model tends to be more confident in predicting a 'canonical' response of the storm track after SSW and strong vortex events (Figure 7): the model successfully predicted the sign of the storm track response in 80% of the 'canonical' events, but only in 25% of the 'non-canonical' events.**

**In response to the comments from the editor and reviewers, we added a new analysis**

to investigate the dynamical aspects that influence the difference between success-
ful versus unsuccessful forecasts, compared to ERA5 (section 3.7). To overcome the
complexity introduced by event-to-event variability, in the new figures 10, 11 and 12,
added in response to the reviewers comments, we focus only events with a 'canonical'
downward response in ERA5. By taking this approach, we are able to estimate how
well the model captures the downward response assuming it is expected to be 'canon-
ical'. We find that ensemble members with a successful prediction of the canonical
downward influence in the Atlantic differ from unsuccessful members mostly in their
representation of tropospheric circulation anomalies after SSW events, indicating that
the troposphere plays a dominant role in a successful prediction of the downward im-
pact of stratospheric anomalies after SSW events.
Following the editors feedback to clarify the temporal evolution of 'canonical' and 'non-
canonical' impacts by weeks, we modified Figure 9 to show for each week whether the
observed response was positive (in "+"; for enhanced cyclone frequency response) or
negative values (in "-"; for reduced response). We believe that these changes help to
improve the manuscript, and provide deeper insights on the biases in capturing the
stratospheric influence on the North Atlantic storm track.

Finally, following the editor's question whether the same factors that determine event-
to-event variability are also responsible for the "unsuccessful" predictions and their
deviation from ERA5, we believe that this is a key question in our field. Our study
shows that removing the inter-event variability, e.g., by choosing only 'canonical'
events, points that the source for unsuccessful prediction can be found both in the
tropospheric and stratospheric anomalies. The insight gained in our study suggest
that capturing the correct persistence of the tropospheric anomalies plays an impor-
tant role in making a "successful prediction" (as shown e.g., by weeks 2-4 of U'850 in
Figure 12). Further work is needed in order to fully investigate the sources of such
forecast errors, and their connection to the external and internal dynamical drivers
that control the downward impact.
Regarding the use of the acronym SVs vs. "strong vortex events" throughout the
manuscript: we primarily use "strong vortex events". The acronym "SV" is used
only in figure titles and legends where an abbreviated name is necessary, but not in
the text.

3. I find it interesting that new Fig 10 shows a stronger polar vortex in forecasts both in SSW
and Strong PV events leading to weaker SSW and even stronger (but displaced) Strong PV in
models. Is that due to, e.g., a vortex bias in models?

As shown in Figure 10, a positive geopotential height anomaly in the lower stratosphere (Z'100) is associated with a weaker stratospheric polar vortex (i.e., normally dominated in winter by a climatological low pressure anomaly over the Northern Hemisphere pole), and a negative anomaly in the lower stratosphere indicates strengthening of the polar vortex. These anomalies represent the downward impact due to the coupling between the stratosphere and the troposphere (during the slow recovery of the stratospheric polar vortex to "normal" winter conditions). We expect the magnitude of these anomalies (i.e., their response to stratospheric forcing) to be partly affected by the model bias.

4. Finally, Fig. 10ab shows that the cyclone mean anomalies in unsuccessful predictions diverge from mean successful forecasts within the first 2-4 days. Perhaps, in addition to 28-day average composites in panels c-j, you can explore what happens during those first days in unsuccessful forecasts? I think this will help further address the question 'why' raised by Reviewer#2 and support your claim that the state of the troposphere is important for successful prediction of SSW. Can you expand on this by, e.g., looking at the evolution of tropospheric jet or other variables that represent the tropospheric circulation?

We follow the suggestion made by Reviewer#2 and the editor to include further analysis on the dynamical aspects associated with storm track predictability after extreme stratospheric events. In the revised version, we expanded the "dynamics" section as follows: Following the reviewers and the editor suggestions, and in addition to 28-day average composites in panels c-j of Figure 10, we explore the time evolution of the state of the troposphere for successful and unsuccessful predictions of the cyclone frequency response after SSW and strong vortex events in the new **Figure 11** and **Figure 12**.
We find that successful predictions after SSW events are characterized by a persistent anomalous pattern between weeks 1 and 4 (this persistent behaviour can be seen both in the mean sea level pressure or the 850-hPa zonal wind pattern). In contrast, unsuccessful predictions are found to be related to various patterns in weeks 1 and 4 (more persistent however between weeks 2 and 3). From this composite analysis, it appears that successful forecasts already exhibit a canonical response in week 1. A similar behaviour was found for strong polar vortex events.

**Minor comments:**

1. Fig.10 caption: I suggest: "Time evolution of cyclone frequency anomaly (in %), zonally averaged over the midlatitude North Atlantic,... Same as (c-f), but for geopotential height anomalies of the 100 hPa surface (Z'100; in gpdm). ... indicated by stippling'

**We have changed the caption as suggested.**

2. Abstract: regarding the mention of health impacts and infrastructure damage due to cyclone activity in Europe, does it refer to the cyclone activity in general or following anomalous polar vortex events? Has it been demonstrated that anomalous polar vortex can be linked to sever wind events? If not, I would remove this statement from the abstract.

**We thank the editor for this comment. We removed the reference to "health impacts and infrastructure damage" as suggested, and rephrased the abstract as follows: "Such changes in the storm track position and associated extratropical cyclone frequency over the North Atlantic and Europe can increase the risk of extreme windstorm, flooding or heavy snowfall over populated regions".**

**2  Reviewer 1:**

**Major comments**

1. Had you considered to split the 28 days period into, for example, two parts (week 1-2 and week 3-4) in the first part of the manuscript? It would probably be interesting to look at least at U850 and cyclone frequency anomalies especially having in mind the results of the predictability part.

**We thank the reviewer for this comment. Following the feedback by the editor and reviewers, we have revised the manuscript and added three new figures (Figure 10,Figure 11, and Figure 12) to demonstrate the time evolution of cyclone frequency response following extreme stratospheric events (SSWs and strong vortex events). This new analysis, especially in figures 11 and 12, analyzes the downward impact from a week by week perspective (as suggested by the reviewer above).**

**In the new figures, we plot the time evolution of cyclone frequency anomalies (ensemble mean) in the North Atlantic Basin (see box in Figure 3) in the period following SSW and strong vortex events, and compare between successful and unsuccessful predictions. First, we also analyze the MSLP and Z'100 anomalies corresponding to these periods. We find that capturing the lower-stratospheric circulation after SSW events is a necessary but not sufficient condition for predicting the downward response (i.e., the greatest difference between successful and unsuccessful prediction is in the troposphere), whereas for strong vortex events capturing both the lower-stratosphere and the troposphere states is necessary.**

**Next, we analyze the time evolution of MSLP and U'850 after successful and unsuccessful predictions. The purpose of this step is to determine the biases in the large-scale circulation associated with successful/unsuccessful forecasts. U850 anomalies,**

in particular, suggest that unsuccessful predictions are also persistent during weeks 2 and 3, but lose their persistence on week 4.

2. The box position choice does not seem well explained. You say that in this region the increase in cyclone frequency is biggest after SSW events (L185), but the anomalies are biggest only in reforecasts (Fig. 3a). Moreover, the anomalies are biggest and statistically significant over the Northern Europe in reanalysis (Fig. 3c), which can be also seen in reforecasts. Had you considered taking a box more to the north-east of its current position? Also, was your choice of the box position based only on the anomalies after the SSW events? I see that the biggest anomalies after the SPV cases are still concentrated inside the box (Fig. 3b), but maybe this can be pointed out in the text.

**The boundaries of the box were determined according to the region of largest increase in cyclone frequency after SSW events (considering all 14 events in the reanalysis dataset). We agree with the reviewer that a larger or shifted box can better capture the anomalies after both SSWs and SPV events. Therefore, we have shifted the southern boundary to 35°N and extended the northern boundary to 55°N. In the revised version, we focus our analysis on the mid-latitude region (35°-55°N) of the North Atlantic (60°W-0°E). This region, located on the southern flank of the North Atlantic storm track, is where the change in cyclone frequency after SSW and strong polar vortex events is the largest. We have updated all the plots in the manuscript according to the new box definition.**

**Minor comments**

L32 "predication" -¿ prediction
**Corrected.**

L34 Add brackets to citation
**Corrected.**

L48-51 This sentence seems to repeat the information given above, please consider removing it or rephrasing the repetition
**We have rephrased that paragraph as suggested by the reviewer to avoid repetition.**

L88 As reforecasts are initialized in conjunction with real-time forecasts, could you provide here the dates/years of the real-time forecasts? You indicate below the model versions used, but this is potentially confusing, as, for example, the 46R1 version does not have reforecasts for December

| Dataset/Forecast | Operation period |
|:---:|:---:|
| Cycle 46R1 | from 11/06/2019 |
| Cycle 47R1 | from 30/06/2020 |
| Cycle 47R2 | from 11/05/2021 |
| Cycle 47R3 | from 13/10/2021 |

Table 1: Implementation dates of each ECMWF model version. Source: `https://confluence.ecmwf.int/display/S2S/ECMWF+Model`.

2019 2.

**The ensemble re-forecasts consist of a 11-member ensemble starting the same day and month as a real-time forecast (Monday and Thursday), covering the past 20 years. The implementation dates of each ECMWF model version is summarized in the table below (Table 1). For example, the reforecasts of December 2, 2019 belongs to the model cycle CY46R1 since it has been computed between 11/06/2019 and 30/06/2020. The reforecast for this date has been initialized on same date as the real-time forecast of 02/01/2020. This reforecast consisted of a 11-member ensemble starting on 2nd January 2000, 2nd January 2001,... to 2nd January 2019 (20 years).**
**To clarify this point, we have added this information to the Methods section (lines 80-85).**

L97 Please consider adding "... in the ECMWF model and in reanalysis..." if you used the same algorithm
**Corrected.**

L100 Consider adding here a remark that the number of the cyclone tracks can be found in Fig.6
**Corrected.**

L104 DJF -¿ DJFM for consistency throughout the text
**Corrected.**

L108 Did you use cross validation when computing the anomalies for each ensemble member?
**Cyclone frequency anomaly for each ensemble member is computed as the difference in the number of cyclones detected in the 28 days after the SSW and the climatological cyclone frequency for this period. As the computation of these anomalies is mathematically straightforward (i.e., an anomaly is defined as a deviation from the climatological mean), we have not performed cross validation when computing the anomalies.**

L108 While I understand the choice of 28 days, it could be better clarified here for better understanding

**We added a clarification in the text regrading the choice of the 28 days period, as follows (lines 110-115):**

**In the NH, anomalies in the tropospheric circulation after extreme stratospheric events (such as SSWs, weak vortex events and strong vortex events) can persist for up to 60 days after their onset (Baldwin and Dunkerton, 2001), and thus may prove to be useful for tropospheric weather prediction. A period of 28 days after the onset of SSWs and strong vortex events is chosen in order to understand the initial tropospheric response and its potential for subseasonal predictions of the surface response.**

L117 I wonder of you checked if there is no difference indeed when using ERA-Interim or ERA-5?

**We have replaced the analysis using ERA-Interim (detection of SSW and SPV events) with ERA-5 data instead. We have updated the manuscript such that the entire analysis is performed using ERA-5 reanalysis. The change does not affect the results of the paper.**

L120 Please specify that the list given in (Butler and Domeisen, 2021) contains only final warming events, rather than all warming events. Or consider omitting this part of the sentence

**We rephrased this part of the sentence to clarify this point.**

L131 Please consider mentioning that the dates of the SSW and SPV can be found later in Figures 7 and 8. I wanted to suggest adding a table with dates, but it seems excessive, since the information appears later in the text.

**A full list of SSW and SPV dates that are used in this study can be found in Figure 7 (and Figure 8). We added this information to the text (line 134).**

Fig 1c The model bias spans from -4.5 to 4.5% while the frequency itself changes from 0 to 45%, do you think that the bias is statistically significant in this case?

**Generally, we find the model bias in cyclone frequency to be statistically significant compared to the reanalysis in most of the North Atlantic sector. In the revised version, we have added an analysis of the statistical significance of the cyclone frequency bias compared to reanalysis in Fig. 1, panels c-d. In these panels, we show that climatological cyclone frequency bias, for model initializations in November to March. We find that the bias is relatively small when considering the first 7-days after the initialization, compared to larger, more statistically significant biases found in the 28-day average. Thus, biases in the range of -4.5 to 4.5% are significant, when evaluated**

**against climatology.**

L154-155 Repetition of "in ERA-5" in the sentence, please remove one of them
**Corrected.**

L155 Did you look at the individual events before constructing the composite? It would be interesting to know which events had stronger response. However, the washed-out signal in the reforecasts might show that the model underestimates the response, especially averaged over 28 days of forecast.
**Indeed, we have examined individual events before constructing the zonal wind composite. We found that some specific events, such as the February 2010 SSW event, exhibit a particularly strong signal in the zonal wind response in the observation.**
**In the revised version, we have uploaded a corrected version of Figure 2 that shows the 850-hPa zonal wind anomalies in all four panels (ERA5 and reforecasts for SSWs and strong vortex events). This change corrects the previous version of Figure 2 in which one of the panels showed the 300-hPa wind instead of 850-hPa. In the revised version of Figure 2, we show that the U850 wind response after SSW and strong vortex events (days 0-28) has a similar magnitude in both the reanalysis and the reforecasts, despite the event-to-event variability. The model also captures the regional extent of the zonal wind response, with significant anomalies in the mid-latitudes of the North Atlantic, as well as in the subtropical Atlantic.**
**Further information on the individual response of SSW and strong vortex events can be found in section 3.5 (Figures 7-9), showing the ensemble response for the North Atlantic domain (Figure 7) and Europe (the new Figure 8).**

L196 "The statistical significance of this shift..." it is not clear whether you refer to the shift compared to all winter days, or the small shift of reforecasts compared to reanalysis
**We refer to statistical significance of the shift in the distribution of cyclone frequency anomalies compared to all winter days. We rephrased this paragraph (lines 205) to clarify this point.**

Fig. 4 As I understand, the figure shows cyclone frequency anomalies after the 14 events in each subplot, but in this case what does the height of the bars show? Counts on y-axis does not add up. If you used a somehow broader statistics, please clarify that in the caption.

**In Figure 4, we analyze the distribution of the cyclone frequency anomaly for each type of stratospheric event (SSW or strong polar vortex event). Anomalies are averaged over a period of 28 days (days 1-28 with respect to the central date of the**

stratospheric event). **Since we have 14 events for each event type, the total number of counts is [N=14] for ERA5 reanalysis and [N=140] for the reforecasts (14 events x 10 ensemble members]). The left y-axis in Figure 4 shows the probability density, hence each bin displays the bin's count, divided by the total number of counts and the bin width, so that the area under the histogram integrates to 1.**
**To address the reviewer's comment, and to provide a clear comparison between "probability density" and "raw counts", a histogram of cyclone frequency anomalies following SSW and strong polar vortex events in terms of raw counts for both reanalysis (panels a,b) and reforecasts (c,d) is shown below (Figure 1). Displaying the probability density allows a direct comparison between the distribution of anomalies in the reanalysis and in the reforecasts (purple bars). Therefore, we have decided to remain with the "probability density". We have clarified this information in Figure 4's caption, and changed the y-axis label from "counts" to "probability density".**

Fig.5 and L212 Could you explain why there are more cyclone tracks (black lines) detected in reforecasts than in reanalysis? I suppose that you used each ensemble member separately rather than ensemble mean, which could be mentioned in text for easier understanding. Also, you mention in Data and Methods that in this part you use more reforecasts from three model versions, but could you explain more in detail why do you use other model versions. The temporal resolution increase to 6-hourly data is understandable here.

**We have added this information to the text to address these important points (lines 212-215). There are more tracks in reforecasts than in reanalysis due to the use of all available ensemble members (11 members) rather than the ensemble mean.**
**Regarding the use of three different model version, this is a result of implementation dates for each new prediction system version (cycle) by the ECMWF. We have included the table with the specific dates (Table 1 in the response letter). Depending on the time in which data has been downloaded, the model version of the reforecast will be different.**

L248 Did you check this correspondence case-by-case here, rather than the overall ratio?
**The ratio of ensemble members with and without a 'canonical' downward response is based on case-by-case examination of the response, as shown in Figures 7-9. In the revised manuscript, this ratio has slightly changed from the previous manuscript due to the new box position in Figure 3. However, we would like to emphasize here that the focus of this manuscript is general behaviour of the downward response in periods following stratospheric events and how well the response is predicted by the ECMWF prediction system. Thus, more emphasis is given to the composite analysis in the first part of the paper. We can easily include the case-to-case analysis of zonal wind and**

[Figure]

Figure 1: Histogram of cyclone frequency anomalies (in counts) following (a,c) SSW and (b,d) strong polar vortex events in ERA5 reanalysis (grey) and in the ECMWF reforecasts (purple). Anomalies are averaged over the mid-latitude North Atlantic. The grey curve in each panel indicates the climatological probability density for all days in DJFM in the reforecasts.

**cyclone frequency in the Appendix, if the reviewers would find it useful.**

Fig.7b,d It would probably be better for understanding if you indicated in the figure that N=10 for enhanced cyclone frequency and N=4 for the reduced, etc.
**We have now added this information to the plot (see x-axis labels).**

L261 Did you have a look why the week-1 hit rate for 11 Feb 2005 was so low, especially considering that the skill is higher on the following weeks and the averaged skill is rather high (0.7 from Fig. 7c)?
**The reasons for the change in skill for the 11 Feb 2005 SSW events are yet unresolved. Analyzing the potential causes requires a more focused study on the dynamics of this specific event, as was done, for instance, for the 2018 SSW event (e.g., Karpechko et al., 2018, Kautz et al., 2020). While this is an interesting question, we believe this analysis is out of the scope of this paper.**

L270 "...predicted a weakening of the cyclone frequency in the period that followed the SSW." As I understand the majority of ensemble members still predicted the increased cyclone frequency on week 1 and 2 in this case, so maybe you can specify that it is not about the period that directly follows the SSW.
**Thanks for this comment, we rephrased this sentence to specify that we refer to week 1 and 2 in this case, and not for the entire period.**

L297 It could be worth specifying that in case of SSW it is about the reduced frequency
**We have rephrased that sentence to clarify that.**

L308 Consider adding "after SSW events in these cases." as temperatures are not always predicted poorly after SSWs.
**We corrected that.**

**3 Reviewer 2**

**3.1 Major comments**

1) First, the study examines only one subseasonal model (ECMWF) and therefore lacks a generalized view of how other leading subseasonal prediction systems reproduce the stratosphere-North Atlantic storm track relationship. The fields from the reforecasts of the other models are readily accessible and possible to be analyzed and compared/contrasted. I am not necessarily advocating using every model, but I think adding a few more will be very useful and strengthen the message.

We thank the reviewer for the comments. Indeed, we agree that a systematic analysis of model biases in the downward impact of extreme stratospheric events across a wide range of subseasonal forecast systems would be an important step towards a better understanding of the role of the stratosphere for prediction of surface climate on subseasonal to seasonal timescales.

However, inter-comparison studies of S2S prediction are way more complex, and require more effort (computationally, as well as time-wise), more data, and usually are done as large community/collaborative studies, as was recently done, for instance, in the Lawrence et al., 2022 for a systematic analysis of model biases in the stratosphere. In the context of extratropical cyclone analysis, this would require implementation of the cyclone detection algorithm for identification and tracking of extratropical cyclones across all prediction systems available in the S2S project.

One of the reasons for using the ECMWF prediction system in this study is due to its more highly resolved stratosphere (relative to other models in the S2S project, e.g., Domeisen et al., 2020). Overall, the ECMWF model has been shown to have a good representation of the variability in the stratospheric polar vortex, in terms of extreme event magnitude and the associated dynamical drivers (Wu et al., 2022). Furthermore, the model represents mid-latitude storm track well, as shown in Fig. 1 in the revised manuscript.

Performing a more complex, extended study on the S2S biases in the prediction of the storm track in the future, may be considered as a follow up on the current manuscript and its main findings using the ECMWF extended-range prediction system.

Please see above also our reply to comment (1) of the Editor.

References:

- Domeisen, D. I., Butler, A. H., Charlton-Perez, A. J., Ayarzagüena, B., Baldwin, M. P., Dunn-Sigouin, E., ... and Taguchi, M. (2020). The role of the stratosphere in subseasonal to seasonal prediction: 2. Predictability arising from stratosphere-troposphere coupling. Journal of Geophysical Research: Atmospheres, 125(2), e2019JD030923. `https://doi.org/10.1029/2019JD030923`

- Lawrence, Z. D., Abalos, M., Ayarzagüena, B., Barriopedro, D., Butler, A. H., Calvo, N., de la Cámara, A., Charlton-Perez, A., Domeisen, D. I. V., Dunn-Sigouin, E., García-Serrano, J., Garfinkel, C. I., Hindley, N. P., Jia, L., Jucker, M., Karpechko, A. Y., Kim, H., Lang, A. L., Lee, S. H., Lin, P., Osman, M., Palmeiro, F. M., Perlwitz, J., Polichtchouk, I., Richter, J. H., Schwartz, C., Son, S.-W., Statnaia, I., Taguchi, M., Tyrrell, N. L., Wright, C. J., and Wu, R. W.-Y.: Quantifying stratospheric biases and identifying their potential sources in subseasonal forecast systems, Weather Clim. Dynam., 3, 977–1001, 2022. `https://doi.org/10.5194/`

```
wcd-3-977-2022
```

- Wu, R. W. Y., Wu, Z., and Domeisen, D. I. (2022). Differences in the sub-seasonal predictability of extreme stratospheric events. Weather and Climate Dynamics, 3(3), 755-776. `https://doi.org/10.5194/wcd-3-755-2022`

2) Next, I found aspects of the methodology confusing. In the methods section, the authors mention that they use the ERA-Interim reanalysis product for determining the state of the polar vortex (strong vs weak) but then use ERA5 for their analyses. Determination of events in the ERA5 dataset is very straightforward. So, to be consistent, the authors should use one reanalysis only throughout their work.

**We thank the reviewer for this comment. We agree with the comment regrading a consistent use of ERA-5 throughout the paper, and therefore we have adjusted the methods section accordingly. In the revised version, the stratospheric extreme events have been detected using the ERA-5 data (which allows a direct comparison with the dates detected by ERA-Interim). Using ERA-5 for events detection does not change the results of this paper.**

3) Next, since the ERA5 is used to initialize the ECMWF reforecasts, and since the two share aspects of their modeling components, independence in the comparisons is hard to justify. Again, this aspect limits the applicability of the results of this work to other forecast systems and reanalyses.

**Although the ECMWF reforecasts are initialized from ERA5 data, they evolve from the reanalysis with time. Therefore, independence in the comparison is not needed for the verification of the forecasts. This study focuses on the ECMWF model from the reasons described in the previous answers, which allows an analysis of the model bias in the 4-weeks following the forecast initialization.**

4) Next, the authors also comment frequently on the limited sample size from ERA5 for their results. This facet factors into their significance testing and other conclusions (e.g., Fig. 5). If sample size is too small, why should we trust the results? I am not saying that the limited sample size is a game-ender for the paper (trust me - this is a constant issue with my own work!). But, to use this concern over and over again in the manuscript as a caveat raises questions as to whether or not the findings are just an artifact of a short sample size.

**We thank the reviewer for pointing out this important topic. Indeed, studies of atmospheric variability, and stratospheric variability in particular, are limited by the small sample size of the observational record. One possible way to increase the sample size is using atmospheric models, that are run for longer time periods compared to the historical record, and thus able to produce a larger number of events (e.g., Afargan-**

Gerstman et al., 2022).

The extent to which we can assess biases is limited by the sample size (e.g., Lawrence et al., 2022; Domeisen et al., 2020). Despite a relatively small sample size (as in this study), these studies are able to assess model biases in the stratosphere across a wide range of subseasonal forecast systems when the results do show evidence of a systematic bias. To overcome the issue of the small sample size in Section 3.7, we take a different approach and combine all events with a canonical surface response (based on a comparison with the reanalysis) and analyze the individual ensemble members (in total 140 members; see the new Figure 10).

Furthermore, we emphasize the variability among the events and its implications on event predictability. Focusing on the inter-variability of these events, given the small sample size limitation, complements the results shown by the composites of zonal wind and cyclone frequency (e.g., Figures 2 and 3) and provides a more detailed perspective on the impact of these events - without making any preliminary assumptions on their systematic bias. In this way, we are able to take a more careful path and overcome the limitation of the historical record.

Finally, in the new Figure 13, we evaluate the uncertainty in the prediction of cyclone frequency after these events, based on the average ensemble spread (the ensemble mean is superimposed in black contours). This additional analysis provides insight regarding the large latitudinal differences in the ensemble spread, suggesting a limited capability of the reforecasts in reproducing the response in mid- and high-latitudes.

References:

- Afargan-Gerstman, H., Jiménez-Esteve, B., and Domeisen, D. I. (2022). On the Relative Importance of Stratospheric and Tropospheric Drivers for the North Atlantic Jet Response to Sudden Stratospheric Warming Events. Journal of Climate, 35(19), 2851-2865.

- Domeisen, D. I., Butler, A. H., Charlton-Perez, A. J., Ayarzagüena, B., Baldwin, M. P., Dunn-Sigouin, E., ... and Taguchi, M. (2020). The role of the stratosphere in subseasonal to seasonal prediction: 2. Predictability arising from stratosphere-troposphere coupling. Journal of Geophysical Research: Atmospheres, 125(2), e2019JD030923. `https://doi.org/10.1029/2019JD030923`

- Lawrence, Z. D., Abalos, M., Ayarzagüena, B., Barriopedro, D., Butler, A. H., Calvo, N., de la Cámara, A., Charlton-Perez, A., Domeisen, D. I. V., Dunn-Sigouin, E., García-Serrano, J., Garfinkel, C. I., Hindley, N. P., Jia, L., Jucker, M., Karpechko, A. Y., Kim, H., Lang, A. L., Lee, S. H., Lin, P., Osman, M., Palmeiro, F. M., Perlwitz, J., Polichtchouk, I., Richter, J. H., Schwartz, C., Son, S.-W., Statnaia, I., Taguchi, M., Tyrrell, N. L., Wright, C. J., and Wu, R.

W.-Y.: Quantifying stratospheric biases and identifying their potential sources in subseasonal forecast systems, Weather Clim. Dynam., 3, 977–1001, 2022. `https://doi.org/10.5194/wcd-3-977-2022`

5) Finally, I was disappointed that the paper did not investigate any physical reasoning for why the storm tracks change as they do in reanalysis vs the reforecasts. The authors mention a few times that their results are "consistent with" previous studies, which is good. But, the reforecasts and their multiple ensemble members offer a fantastic opportunity for the authors to address the "why." They could explore changes in wave fluxes, baroclinicity, jet stream dynamics, etc. and provide an idea of why the stratosphere is influencing the storm tracks the way it is. I think this is a missed opportunity with this paper, thus making it contribution less novel than it otherwise could be.

**We thank the reviewer for these suggestions. We have made substantial changes to the manuscript to address the physical aspect of the storm track change in the reforecasts, compared to the reanalysis. Specifically, we have added a new subsection (3.7) to analyze and discuss the dynamical aspects of successful and unsuccessful predictions after SSW and strong vortex events. For this purpose, we explore the changes in troposphere, represented by the mean sea level pressure (MSLP) anomaly, and in the lower stratosphere, represented by the geopotential height anomaly at 100 hPa (Z'100) in the new Figures 10, 11 and 12.**

**Predictability of the downward impact after extreme stratospheric events strongly differs among events, even of the same type (e.g., Domeisen et al., 2020; Wu et al., 2022). The reasons for the observed differences in the predictability are not yet resolved, and often require an analysis from a case-by-case perspective, as was done for example for the 2018 SSW events (e.g., Karpechko et al., 2018, Kautz et al., 2020).**

**To determine the source of predictability of the downward response, we analyze the physical difference between SSWs that were "successfully"/"unsuccessfully" predicted (based on a criterion for a successful prediction, defined as forecasts in which the majority of ensemble members predict the observed sign of response in the midlatitude North Atlantic box). To guarantee a consistent surface response, only events with a canonical downward response are analyzed. We find that ensemble members with a successful prediction of the canonical downward influence after SSW event differ from unsuccessful members mostly in their representation of tropospheric circulation anomalies after SSW events: the unsuccessful members do not predict the North-South dipole pattern in the Atlantic that corresponds to a negative NAO pattern (as shown by the MSLP patterns in Figure 10), despite well capturing the circulation anomalies in the lower stratosphere (as shown by the Z'100 patterns in Figure 10). These results indicate that the troposphere plays a dominant role in the downward**

impact of stratospheric anomalies after SSW events. Following strong polar vortex events, however, members with successful predictions differ from unsuccessful members in both their tropospheric and lower stratospheric anomalies.

Overall, this analysis sheds light on our conclusions presented in the manuscript, regarding the role of the tropospheric circulation in determining the predictability of the downward response of extreme stratospheric events. We have added this analysis to the manuscript (Section 3.7; Figures 10, 11 and 12).

References:

- Domeisen, D. I., Butler, A. H., Charlton-Perez, A. J., Ayarzagüena, B., Baldwin, M. P., Dunn-Sigouin, E., ... and Taguchi, M. (2020). The role of the stratosphere in subseasonal to seasonal prediction: 2. Predictability arising from stratosphere-troposphere coupling. Journal of Geophysical Research: Atmospheres, 125(2), e2019JD030923. `https://doi.org/10.1029/2019JD030923`

- Karpechko, A. Y., Charlton-Perez, A., Balmaseda, M., Tyrrell, N., and Vitart, F. (2018). Predicting sudden stratospheric warming 2018 and its climate impacts with a multimodel ensemble. Geophysical Research Letters, 45(24), 13-538. `https://doi.org/10.1029/2018GL081091`.

- Kautz, L. A., Polichtchouk, I., Birner, T., Garny, H., and Pinto, J. G. (2020). Enhanced extended-range predictability of the 2018 late-winter Eurasian cold spell due to the stratosphere. Quarterly Journal of the Royal Meteorological Society, 146(727), 1040-1055. Wu, R. W. Y., Wu, Z., and Domeisen, D. I. (2022). Differences in the sub-seasonal predictability of extreme stratospheric events. Weather and Climate Dynamics, 3(3), 755-776. `https://doi.org/10.5194/wcd-3-755-2022`

**Other Comments**

1. The acronym "SPV." The use of this acronym is confusing - it is normally used to mean "stratospheric polar vortex" in many other papers. Furthermore, I don't find that the acronym is necessary in the work - "strong vortex events" is clear enough and not overly long. I recommend that the authors reconsider using this acronym.

**The use of the acronym "SPV" for "strong polar vortex" can be found in the literature (e.g., Oehrlein et al., 2020, Díaz-Durán et al., 2017). However, to avoid confusion due to the different uses of this acronym, we corrected "SPV" to "strong vortex event" throughout the manuscript. In specific places we use the acronym "SV" for "strong vortex".**

- Oehrlein, J., Chiodo, G., and Polvani, L. M. (2020). The effect of interactive ozone chemistry on weak and strong stratospheric polar vortex events. Atmospheric Chemistry and Physics, 20(17), 10531-10544.

- Díaz-Durán, A., Serrano, E., Ayarzagüena, B., Abalos, M., and de la Cámara, A. (2017). Intra-seasonal variability of extreme boreal stratospheric polar vortex events and their precursors. Climate Dynamics, 49, 3473-3491.

2. Lines 154-155. I don't understand this sentence. How is the "response in ERA5"... "stronger in ERA5?"
**We rephrased this paragraph and removed this sentence.**

3. Line 221. Either the results are statistically significant or they are not - they cannot be "partly significant."
**Thank you for pointing this out. We have rephrased that sentence to clarify that the results in that case (difference in cyclone intensity) are not significant in ERA5 (lines 222-223).**

4. Lines 281-283. Is this a "result" or "finding" that is unique to this work? I think that finding has already been shown in many past works and is also based on the fundamentals of what the jet stream is.
**We rephrased this paragraph to emphasize that our analysis is consistent with previous studies, in context of the expected stratospheric impact. However, we emphasize a possible overconfidence of the model with respect to reanalysis to predict the canonical response after SSW events - a topic which has received less attention in the literature.**

5. Lines 291-293. How would the authors propose to increase the sample size to meet their objective of determining the robustness of the results? (See my comments above as well.)
**One possibility for increasing the sample size is to use a modeling study, hence to run a model for a longer time and generate more SSW events (e.g., Afargan-Gerstman et al., 2022). However, such procedure is less relevant when analyzing the ECMWF forecasts, as done in the current study. To overcome this issue, we take a different approach in Section 3.7, and perform our analysis on the individual ensemble members (140 members for each event type) rather than an event-based analysis (14 events for each event type). By taking this approach, we are able to analyze larger sample size for each event type, and to gain more statistical insights on their predictability and hence dynamical aspects.**

6. Figures 2 and 3. How is significance tested exactly for the reforecasts? What is the null hypothesis?

**In Figures 2 and 3 significance is tested based on a Student's t-test. The null hypothesis (H0) is there is no difference between the means of these two variables (i.e., zonal wind anomalies and 0). Significance is tested for each grid point. An additional and more detailed significance testing is performed in section 3.4, where we investigate how the average cyclone life cycle characteristics depend on the extreme states of the stratospheric polar vortex. In Figure 5 and Figure 6, the confidence interval is obtained from a bootstrapped distribution of median latitudes (based on 1000 random resamples of the tracks with replacement).**

7. Figure 7. "Successfully" is spelled incorrectly in the y-axis labels of panels (b) and (d). Also, it is unclear what an "increase of cyclone frequency anomaly" means. Is it that it is a positive anomaly, or that the anomaly actually gets more positive over some time?

**We have corrected the typo. As for the meaning of "increase of cyclone frequency anomaly", this term refers to a positive anomaly in cyclone frequency. Red bars in Fig. 7a indicate the proportion of ensemble members that show an average increase in cyclone frequency over the selected region, whereas blue bars indicate a decrease.**

8. Code and data availability. The authors have not provided a public-accessible repository where their code is available. Please set up a Github and place your code on there for transparency and accessibility.

**We thank the reviewer for this comment. We have created a public-accessible Github repository for the code and datasets (`https://github.com/hillaag/downward_impact_analysis_tools_for_S2S.git`). We now specify this information under the Code Availability section.**

---

## Author Response (AR3)

**Response to Reviewers - 2nd revision**

Dear Editor,

We would like to thank all reviewers and the editor for their reviews of our manuscript and their insightful comments. Please find our detailed responses to the reviewers' comments and suggestions below. The changes have been included into the manuscript (indicated in **bold**). All line numbers refer to the new (annotated) version of the manuscript.

Sincerely,

The authors

**Reviewer 1**

**Minor comments:**

Figure 3 There are currently two black boxes which is confusing throughout the text. I suggest distinguishing between them, for example, using another color or a dashed line for one of them.
**We thank the reviewer for their comments. We changed the color of the European box to red. The North Atlantic box is marked in black.**

L198 As you now have boxes plotted in all of the subplots it's enough to say 'Figure 3'. However, it is good to specify here which box you're referring to (see my comment above).
**We added multiple specification of which box is refereed to.**

Figure 8 caption: although you mention the coordinates, it's better to specify here which box you're referring to once you've distinguished the boxes (see my comment above).
**Corrected**.

Figure 10 Please mention in the caption that this is a 28d average
**We added this information to the caption.**

L342 Although you mention using Z100 in the Data and Methods, I believe that this is the first time you mention Z'100 (Z100 anomalies) so it makes sense to specify it here.
**We added an explanation for Z'100 acronym in section 3.7 and removed any unused acronym from the Data and Methods section.**

L358 please fill in the Figure number instead of '??'
**We corrected the missing figure reference.**

L370 typo 'to be associated'
**Corrected.**

L373 probably 'to SSW events'
**Corrected.**

L452 I think that you forgot to add the link to Github in the Code availability section.
**Thank you for this comment. The previous version inadvertently omitted the link to the Github site. We have updated the "Code availability" section with the correct link.**

**Reviewer 2**

**Overall Opinion:**

This is my second review of the manuscript. The prior version of the manuscript lacked two key elements that I thought made it unsuitable for publication: (1) The study focused on only one model, whereas the database has several other models with hindcasts that could be explored for the same phenomenon; and (2) the paper lacked any dynamical insight into why the results were the way they were. The authors responded that doing the analysis for more than one model would be too laborious and that the ECMWF model was a well-trusted subseasonal model. I kind of agree with this thought, though I think the paper would be strengthened with more than one model analyzed for the study. But, I won't hold up publication of the study based just on that issue. For the second point, the authors expanded their analysis to look at the nature of downward propagation and the tropospheric circulation after a strong or weak vortex event. Specifically, the authors contrast between ensemble members which correctly predict the anomaly of the North Atlantic cyclone frequency following a strong/weak polar vortex event and those that do not. However, I don't think the methodology used actually addresses the dynamical interpretation of the results that the authors intend.

So, taken together, I would consider the paper ready for publication but only after major revisions, particularly to the dynamical interpretation portion of the paper.

**Major comments:**

1. Sampling Issues and Figure 4. The authors continue to note that sample size is a concern for many of their results, and I agree with this point. However, one area this is not addressed enough is in Figure 4. In particular, the sample size between reanalysis and reforests for the SSW and strong vortex events is about a factor of 10 different, which makes comparison of the probability distribution functions (PDFs) very difficult (in fact, I question the reproducibility and representation of a distribution of a variable with only 14 samples). For example, the authors could repeatedly

sample 14 random cases (with replacement) from the reforecasts to make a comparative PDF with reanalysis. I think in its current form, it is hard to argue about statistically significant differences in these distributions, whether comparing the cases or comparing reanalysis to reforecasts. The authors may want to pursue alternate strategies to strengthen this argument in the paper.

**We thank the reviewer for this comment. Figure 4 shows distributions of cyclone frequency anomaly in the central North Atlantic for reanalysis and re-forecasts for SSW and strong vortex events. A statistical significance test, a two-sample Kolmogorov-Smirnov test, has been performed between two fitted distributions. However, we are aware that it is a difficult task to do a comparison of the two distributions given their different sample size (14 for the reanalysis samples, and 140 for the probabilistic forecasts). Therefore, and given the current structure of the manuscript, Figure 4 has been removed from the manuscript. Instead of further statistical comparison between reanalysis and the reforecasts, in the revised manuscript we extended the discussion on the dynamical causes for successful or unsuccessful prediction of the canonical surface response after stratospheric events (section 3.7) using a relatively larger sample size (100 samples) provided by the reforecasts. One of the new additions includes a new figure (Figure 11) added to this subsection.**

2. Dynamical Interpretation. I like the efforts that the authors made in trying to bring some dynamical insight into their model evaluation study of Atlantic storm track changes due to stratospheric polar vortex variability. However, I am not convinced that the analyses shown actually accomplish this effort. In particular, the authors use mean sea level pressure (MSLP) to represent the tropospheric circulation changes after "successful" and "unsuccessful" forecasts. However, storm tracks are defined in this study using MSLP. So, it is a bit circular to argue that differences in MSLP ("the tropospheric circulation") are the leading reason why there are changes in storm tracks (which are determined by MSLP). I like the use of lower tropospheric winds and even looking at the lower stratosphere (Z100). But, the authors should reconsider how they measure the tropospheric circulation and consider other variables for that other than MSLP.

**We thank the reviewer for this comment. MSLP anomalies, shown originally in figure 11, represent surface circulation anomalies and provide different insights than the cyclone frequency. In particular, MSLP anomalies consist of both the high and the low pressure anomalies of the synoptic-scale flow, while cyclone frequency is primarily associated with low pressure systems (in the Northern Hemisphere). Thus, MSLP and cyclone frequency can represent different quantities.**
**Following the reviewer's suggestion, we changed the variable that we use to represent tropospheric circulation, and we now use zonal wind at 850 hPa (U'850). We added an analysis of U'850 in successful/unsuccessful forecasts (Figure 9). In addition, we**

replaced MSLP with U'850 in an analysis of the time evolution of lower tropospheric anomalies (figure 10), thus using the same variable to represent tropospheric circulation anomalies in both figures.

**Minor comments:**

1. Lines 14-18. This sentence is long and confusing to understand. Please revise.
**We revised this sentence in the Abstract, as follows:**
**"However, although the response of cyclone frequency following SSWs with a canonical surface impact is typically well-captured during weeks 1-4, less than 25% of the reforecasts manage to capture the response following SSWs with a 'non-canonical' impact. This suggests a possible overconfidence in the reforecasts with respect to reanalysis in predicting the canonical response after SSWs, although it only occurs in about two thirds of the events".**

2. Lines 74-75. The acronym "ERA5" already contains the word "reanalysis" in it. So, it is redundant to say "ERA5 reanalysis."
**Corrected. We replaced "ERA5 reanalysis" with "ERA5" throughout the manuscript.**

3. Line 228. Please move the comma from after "ERA5" to after "reforecasts."
**Corrected.**

4. Lines 313. There is no need to define "MSLP" again here. Also, I think it is unnecessary to introduce another acronym into the paper for the 100 hPa geopotential height anomalies. Instead, the authors can just use the already-defined acronym for 100 hPa geopotential height anomalies (Z100). Can you just write "Z100 anomalies?"
**We removed the repeated MSLP acronym in line 313, and in the Methods section (line 75) we removed other acronym (U10, U850 and Z100) that are not used throughout the revised manuscript. Instead, we added acronym to U'850 and Z'100 as these are used multiple times in section 3.7.**

5. Lines 338-340. This sentence structure (with the parentheses) is no longer favored in journal articles for readability and understanding. Please rephrase as two sentences or in another way. Same comment for Lines 411-412.
**We thank the reviewer for this clarification. We rephrased both of these sentences to improve their readability and understanding.**

6. Line 353. It looks like there is a missing figure reference here ("??").

**We corrected the missing figure reference.**

7. Lines 362-363. I am unclear what "larger natural variability" means, particularly in reference to model reforecasts. What does "natural variability" in a simulated atmosphere mean?
**Originally, the term natural variability in this context referred to variation in climate parameters of the simulated atmosphere caused by nonhuman forces. To clarify this meaning, we replaced the term "larger natural variability" by "larger variability".**

8. Line 364. "...found to b associated with a..." —¿ "...found to be associated with a..."
**Corrected.**

9. Lines 445-448. The authors previously mentioned that they were going to have a Github site to make their data publicly accessible. This site is not listed here - please add the information for completion.
**Thank you for this comment. The previous version inadvertently omitted the link to the Github site. We have updated the "Code availability" section with the correct link.**